# UniteFormer: Unifying Node and Edge Modalities in Transformers for Vehicle Routing Problems

**Dian Meng**[1,5]      **Zhiguang Cao**[2]      **Jie Gao**[3]      **Yaoxin Wu**[4]      **Yaqing Hou**[1,5*]

[1]School of Computer Science and Technology, Dalian University of Technology (DUT)
[2]School of Computing and Information Systems, Singapore Management University
[3]Department of Transport and Planning, Delft University of Technology
[4]Department of Industrial Engineering and Innovation Sciences, Eindhoven University of Technology
[5]Key Laboratory of Social Computing and Cognitive Intelligence (DUT), Ministry of Education, China
mengdian@mail.dlut.edu.cn, zhiguangcao@outlook.com, J.Gao-1@tudelft.nl,
wyxacc@hotmail.com, houyq@dlut.edu.cn

## Abstract

Neural solvers for the Vehicle Routing Problem (VRP) have typically relied on either node or edge inputs, limiting their flexibility and generalization in real-world scenarios. We propose *UniteFormer*, a unified neural solver that supports node-only, edge-only, and hybrid input types through a single model trained via joint edge-node modalities. UniteFormer introduces: (1) a *mixed encoder* that integrates graph convolutional networks and attention mechanisms to collaboratively process node and edge features, capturing cross-modal interactions between them; and (2) a *parallel decoder* enhanced with query mapping and a feed-forward layer for improved representation. The model is trained with REINFORCE by randomly sampling input types across batches. Experiments on the Traveling Salesman Problem (TSP) and Capacitated Vehicle Routing Problem (CVRP) demonstrate that UniteFormer achieves state-of-the-art performance and generalizes effectively to TSPLib and CVRPLib instances. These results underscore UniteFormer's ability to handle diverse input modalities and its strong potential to improve performance across various VRP tasks.

## 1   Introduction

Vehicle Routing Problems (VRPs) are fundamental in logistics [21], navigation systems [12], and drone delivery [45], with significant theoretical and practical relevance. Recent advances have seen increasing interest in deep learning-based neural solvers for VRPs, offering strong generalization and improved computational efficiency over traditional exact and heuristic algorithms [14, 3, 4]. These methods include both autoregressive models that learn construction policies from data [24, 10, 31], as well as learning-based improvement solvers that enhance classical optimization procedures. However, many of these models, particularly construction-based ones, make an overly simplifying assumption: they rely solely on either node coordinates or edge distances as input. This leads to several limitations. First, training separate models for each input modality (node or edge) is inflexible and impractical for real-world applications. Second, switching between different input types requires retraining from scratch, incurring substantial computational costs. Third, such single-modality training neglects the complementary information between node and edge inputs, preventing the model from learning transferable features and reducing its capacity to discover high-quality solutions.

---

[*]Corresponding author.

39th Conference on Neural Information Processing Systems (NeurIPS 2025).

We argue that hybrid training with both node and edge information offers a more general and informative representation of the problem. While existing methods train only on a single modality, our approach allows joint encoding and interaction across modalities, leading to better-informed policies and improved solution quality. To this end, we propose *UniteFormer*, a unified neural solver for VRPs that supports hybrid training and generalizes across input modalities. Unlike conventional solvers, UniteFormer is trained once and can handle node-only, edge-only, or mixed edge-node inputs without retraining. This makes it more flexible and applicable to diverse real-world scenarios.

Specifically, UniteFormer consists of a mixed encoder and a parallel-attention decoder. The mixed encoder includes two sub-encoders: edge-aware sub-encoder and node-focused sub-encoder. The edge-aware sub-encoder integrates residual gated graph convolutional networks (GCNs) with self-attention to jointly process node and edge features, facilitating cross-modality interaction. The node-focused sub-encoder encodes node features independently using attention mechanisms, thereby further enhancing the ability to encode node information. Together, they produce rich global embeddings that capture complementary structural information. The decoder features a parallel architecture and nonlinear query mechanisms, incorporating query mapping and a feed-forward (FF) layer to enhance its representational capacity. Our contributions are outlined as follows:

- We present UniteFormer, the first unified neural solver capable of solving VRPs with node-only, edge-only, or hybrid inputs using a single trained model.
- We introduce a novel mixed encoder that combines residual gated GCNs with attention mechanisms, which can effectively and jointly process node and edge features to capture cross-modal interactions between them.
- We design a decoder with a parallel-attention architecture and nonlinear query mechanisms, which enhance the expressiveness of the policy network.
- Experiments on TSP and CVRP with all three input types show that UniteFormer achieves state-of-the-art results. It also generalizes well to real-world TSPLib and CVRPLib benchmarks, and supports applications such as the Asymmetric TSP (Appendix F).

## 2   Related Work

**Modality-Specific Neural Solvers for VRPs.**   Neural approaches have emerged as powerful alternatives for solving VRPs by leveraging advances in deep learning and neural combinatorial optimization [3, 35, 32, 47, 26, 27, 17]. The introduction of pointer networks [42] and the Transformer architecture [40] laid the foundation for early neural VRP solvers such as in [2] and [34].

**1) Node-based models:** Most existing neural solvers focus on node coordinate inputs. Notable examples include AM [22], POMO [24], and Sym-NCO [19], which significantly improved solution quality for classical VRPs. More recent works have advanced training strategies. For instance, Bdeir et al. [1], Drakulic et al. [10], and Luo et al. [31] applied dynamic input re-encoding during training to enhance generalization. Among them, Drakulic et al. [10] introduced Bisimulation Quotienting (BQ) to reformulate the MDP for more robust generalization. Luo et al. [31] proposed a light encoder heavy decoder (LEHD) model trained via supervised learning on partially reconstructed 100-node instances. These methods are all fundamentally built on node coordinate inputs, capturing spatial structure through positional embeddings.

**2) Edge-based models:** Edge-centric models are a more recent development. Kwon et al. [23] introduced MatNet, a matrix encoding network that operates on pairwise distance matrices, and demonstrated strong performance on the asymmetric traveling salesman (ATSP) and flexible flow shop (FFSP) problems. Lischka et al. [28] proposed GREAT, a sparse graph edge attention model that constructs high-quality solutions by exploiting sparse edge relationships. Building on this, Meng et al. [33] proposed an efficient edge-based EFormer, which further extends and optimizes edge-based problems and achieves excellent results on the TSP and CVRP.

**3) Hybrid edge-node models:** A smaller body of work explores models that jointly use node and edge information. Joshi et al. [18] proposed a GCN-based edge probability predictor that uses both node coordinates and edge weights to guide beam search. Wang et al. [44] developed a distance-aware reshaping method (DAR) that biases attention mechanisms using Euclidean distances. Zhou et al. [49] introduced an instance-conditional adaptive model (ICAM) that integrates both node and edge features to improve adaptability across instance sizes.

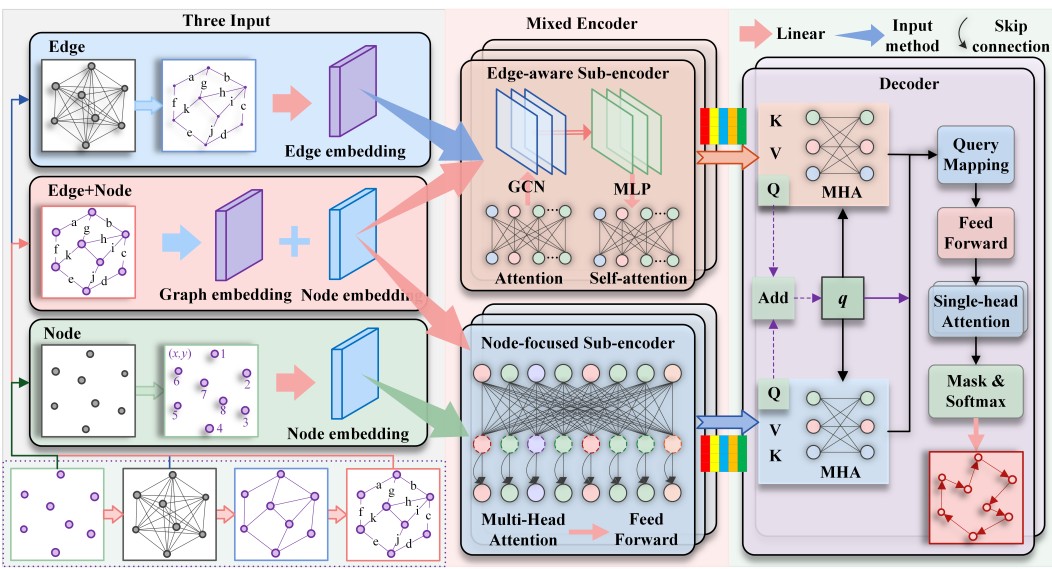

Figure 1: Overview of the UniteFormer framework.

**Unified Neural Solvers for VRPs.** Unified models that generalize across VRP variants have gained attention due to their versatility and practicality. Wang and Yu [43] proposed a multi-task neural solver using a multi-armed bandit framework to train across multiple combinatorial optimization problems. Drakulic et al. [9] introduced GOAL, a supervised learning-based agent capable of solving diverse COPs. Within the VRP domain, Ruis et al. [37] used attribute composition for zero-shot generalization across multiple VRP variants. Liu et al. [29] extended the reinforcement learning-based POMO model to a multi-task setting (POMO-MTL), and Zhou et al. [50] further proposed MVMOE, a mixture-of-experts model to improve generalization. Building on this, Liu et al.. [30] proposed a curvature-aware pre-training framework that effectively improved their performance. Federico et al. [5] introduced RouteFinder, a modular baseline framework for VRP variant modeling. Our work aligns with this line of research but focuses on input-modality unification rather than task-level generalization. UniteFormer is the first model that simultaneously processes VRPs defined by node, edge, or hybrid representations in a single framework, offering strong generalization, improved efficiency, and broad applicability to real-world VRPs.

## 3 UniteFormer

Transformer-based neural VRP solvers typically adopt light decoder architectures [22, 24], where the decoder uses static node embeddings as keys and values throughout the attention layers. In contrast, we replace these static embeddings with two context-aware embeddings that encode both edge relationships and node coordinates, and process them in parallel within the decoder. To effectively encode heterogeneous input modalities, we introduce a novel mixed encoder architecture that combines residual gated GCNs with attention mechanisms. The mixed encoder includes two sub-encoders: an edge-aware sub-encoder and a node-focused sub-encoder, collaboratively processing node and edge features to capture cross-modal interactions between them. In addition, we enhance the parallel-attention decoder with query mapping and a feed-forward layer to form our proposed **UniteFormer**. The overall architecture of UniteFormer is illustrated in Figure 1. In the following, we first present three input modalities in UniteFormer, then introduce the two sub-encoders of the mixed encoder in detail, and finally report the specific implementation of the decoder.

### 3.1 Input Modalities in UniteFormer

A VRP instance is defined over a graph $G = \{X, E\}$, where $X = \{x_i\}_{i=0}^{N}$ denotes the nodes (with $x_0$ as the depot), and $e(x_i, x_j) \in E$ represents the edge between nodes $x_i$ and $x_j$. UniteFormer

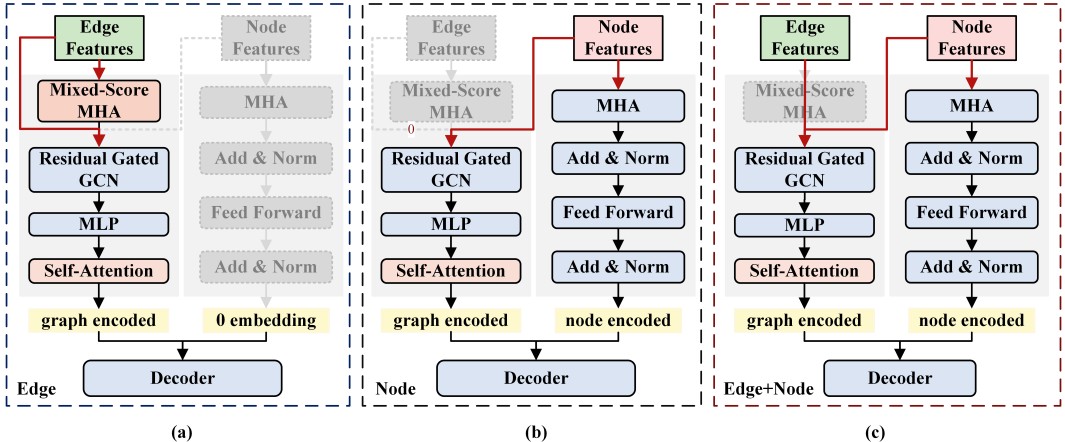

Figure 2: Three input modalities for UniteFormer: (a) Edge-only input. (b) Node-only input. (c) Hybrid input. The Mixed Encoder includes: Edge-aware (left) and Node-focused Sub-encoder (right).

supports three input modalities: *edge-only input*, where only edge weights $E$ are provided; *node-only input*, where only node coordinates $X$ are provided; *hybrid input*, where both node coordinates $X$ and edge weights $E$ are available.

The three input configurations are shown in Figure 2. Different input types activate different branches of the edge-aware sub-encoder (left) and the node-focused sub-encoder (right). A single unified model is trained, with encoder components dynamically adapted to each input type. Specifically, **Edge-only input** (Figure 2(a)): The node-focused sub-encoder is disabled (i.e., replaced with a zero embedding), and only the edge-aware sub-encoder processes edge features; **Node-only input** (Figure 2(b)): Without edge weights, node features are passed to the GCNs directly, while the edge features are set to zero embeddings. **Hybrid input** (Figure 2(c)): Both node and edge features are used. The mixed-score Multi-Head Attention (mixed-score MHA) is bypassed, and the GCNs receive the raw edge and node features.

### 3.2 Edge-aware Sub-encoder: Efficient Fusion of GCNs and Attention Mechanisms

To simultaneously capture edge and node information, we propose a novel sub-encoder architecture, i.e., the Edge-aware Sub-encoder, that integrates GCNs with attention mechanisms. When edge information is provided, we apply a mixed-score MHA block to derive intermediate node features. Across all three input modalities, we then apply residual gated GCNs to jointly process edge and node features in a unified representation space. Finally, we introduce an additional self-attention layer, which proves especially effective in edge-based settings.

**Mixed-Score Attention Layer.** Inspired by [23], we adopt a multi-head mixed-score attention mechanism to encode the edge weight matrix. This module follows the structure of standard Transformer attention [40], but replaces the traditional scaled dot-product computation with a mixed-score formulation (see Appendix A). Inputs to this block include: a zero vector $h_0$, a randomly selected one-hot vector $h_v$ from a predefined pool, and the edge weight matrix $D_{ij}$. This setup allows dynamic embedding generation and facilitates instance-level augmentation by feeding the same instance multiple times with varying vector combinations. The zero vector can optionally be replaced by another one-hot vector, though we default to using the zero vector. The encoded relation matrix $h_i^{(P)}$ is computed only when edge inputs are present:

$$\hat{h_i}^{(P)} = \text{NORM}(h_0 + \text{mixed-scoreMHA}(h_0, h_v, D_{ij})), \qquad (1)$$

$$h_i^{(P)} = \text{NORM}(\hat{h_i}^{(P)} + \text{FF}(\hat{h_i}^{(P)})), \qquad (2)$$

where $\text{mixed-scoreMHA}(\cdot)$ denotes the mixed-score attention layer, $\text{FF}(\cdot)$ is a feed-forward network with one hidden layer and ReLU activation, and $\text{NORM}(\cdot)$ denotes batch normalization [16].

**Residual Gated Graph Convolution Layer.** We next feed node and edge features into residual gated GCNs. Node coordinates $x_i$ are embedded as $h$-dimensional vectors. The edge weight matrix $D_{ij}$ and the edge adjacency matrix $\Theta_{ij}^{\mathrm{knn}}$ are embedded as $\frac{h}{2}$-dimensional vectors:

$$\alpha_i = \omega_1 x_i + b_1, \tag{3}$$

$$\beta_{ij} = \omega_2 D_{ij} + b_2 || \omega_3 \cdot \Theta_{ij}^{\mathrm{knn}}, \tag{4}$$

where $\omega_1 \in \mathbb{R}^h$, $\omega_2, \omega_3 \in \mathbb{R}^{\frac{h}{2}}$, $b_1$, $b_2$ are biases, and $||$ denotes vector concatenation. The input node and edge embeddings for the GCN are adaptively initialized according to input types. Particularly, 1) Edge-only: $y_i^0 = h_i^{(P)}$, $e_{ij}^0 = \beta_{ij}$; 2) Node-only: $y_i^0 = \alpha_i$, $e_{ij}^0 = h_0$; 3) Edge and Node: $y_i^0 = \alpha_i$, $e_{ij}^0 = \beta_{ij}$. We denote node and edge embeddings at layer $l$ as $y_i^l$ and $e_{ij}^l$, respectively. Following [6], we apply ReLU activation and residual connections to obtain the next-layer embeddings:

$$y_i^{l+1} = y_i^l + \mathrm{ReLU}(\mathrm{NORM}(W_1^l y_i^l + \phi_{ij}^l \odot W_2^l y_j^l)), \text{ with } \phi_{ij}^l = \sum_{j \sim i} \frac{\sigma(e_{ij}^l)}{\sum_{j' \sim i} \sigma(e_{ij'}^l) + \xi}, \tag{5}$$

$$e_{ij}^{l+1} = e_{ij}^l + \mathrm{ReLU}(\mathrm{NORM}(W_3^l e_{ij}^l + W_4^l y_i^l + W_5^l y_j^l)), \tag{6}$$

where $W_*^l$ are learnable weight matrices, $\sigma$ is the sigmoid function, $\xi$ is a small constant for numerical stability, and $\odot$ denotes element-wise multiplication. This formulation enables anisotropic information diffusion on graphs by incorporating learned edge attention maps $\phi_{ij}^l$. To further process the node embeddings, we apply the MLP to the GCN output $y_i^l$, yielding values $m_i^l = \mathrm{MLP}(y_i^l)$ constrained to $[0,1]^2$.

**Self-Attention Layer.** To enhance the model's capacity for global context encoding, we introduce an additional self-attention layer after the MLP. This is particularly important in the edge-only setting, where the node-focused sub-encoder is disabled. In such cases, this layer significantly improves the model's ability to propagate and transform information across the graph:

$$h_L^M = \text{self-attention}(m^L), \tag{7}$$

where $m^L$ is the MLP output, and $h_L^M$ is the final output of the edge-aware sub-encoder. The detailed computational process in self-attention layer is provided in Appendix A.

## 3.3 Node-focused Sub-encoder: Expressive Encoding of Node Information

The edge-aware sub-encoder, which is built on GCNs and augmented with a self-attention layer for encoding, offers a significant boost for edge-based input. However, it falls short in handling node features compared to the conventional encoder mechanism [24]. Therefore, we introduce the classic attention mechanism (i.e., Node-focused Sub-encoder) to make up for this deficiency, which can effectively improve the performance of node-based input. The node-focused sub-encoder consists of $L$ stacked layers, each comprising two sublayers: a multi-Head attention (MHA) sublayer and a feed-forward (FF) sublayer. Each sublayer incorporates residual connections [13] and layer normalization [16]. Let $h_i^{(l)}$ denote the embedding of node $i$ at layer $l$, and let $H^{(l)} = \{h_1^{(l)}, h_2^{(l)}, \ldots, h_n^{(l)}\}$ represent the node embeddings at layer $l$. The forward computation at the $l$-th layer is given by:

$$\hat{h}_i^{(l)} = \mathrm{NORM}\left(h_i^{(l-1)} + \mathrm{MHA}\left(h_i^{(l-1)}, H^{(l-1)}\right)\right), \tag{8}$$

$$h_i^{(l)} = \mathrm{NORM}\left(\hat{h}_i^{(l)} + \mathrm{FF}\left(\hat{h}_i^{(l)}\right)\right), \tag{9}$$

where $\mathrm{MHA}(\cdot)$ denotes the multi-head attention, $\mathrm{FF}(\cdot)$ is a feed-forward network, and $\mathrm{NORM}(\cdot)$ applies layer normalization. This structure allows the sub-encoder to capture complex dependencies between nodes in a permutation-invariant manner. $H^{(L)}$ represents the final output of the $L$-th attention layer. Specifically, when the input consists of edges only, the sub-encoder is deactivated, and its output is set as $h_L^N = h^{(0)}$; when the input includes nodes, the output is taken as $h_L^N = H^{(L)}$.

## 3.4 Decoder: Parallel-Attention Decoding with Enhanced Query Representation

A critical component of the decoder is the context query vector $q$, which is used to compute attention scores over node embeddings and generate the probability distribution for the next node. In prior

works, $q$ is often constructed as a linear combination of node embeddings, which limits its representational capacity due to its inherent linearity [15]. To better capture contextual dependencies, we design a decoder architecture with two key enhancements: (1) a *parallel-attention architecture* that separately computes attention scores using two sets of encoded embeddings, and (2) a *nonlinear query mechanism* that increases the expressive power of the query vector. Specifically, we apply query mapping and a feed-forward network with residual connections following the MHA layers.

**Parallel-Attention Decoding.** The edge-aware sub-encoder and node-focused sub-encoder produce two global embeddings, denoted as $h_L^M$ and $h_L^N$, where the superscripts $M$ and $N$ indicate the edge-aware and node-focused encoding paths, respectively. During decoding, these embeddings are processed in parallel to obtain the decoder context vectors at decoding step $t$, given by $H_c^M = [h_1^M, h_t^M]$ and $H_c^N = [h_1^N, h_t^N]$. These vectors are used to form temporary queries:

$$q^M = W_1^M h_1^M + W_2^M h_t^M, \tag{10}$$

$$q^N = W_1^N h_1^N + W_2^N h_t^N, \tag{11}$$

$$q = q^M + q^N, \tag{12}$$

where $W_1^M, W_2^M, W_1^N$ and $W_2^N$ are learnable matrices that transform the start node embeddings $h_1^M, h_1^N$ and current node embeddings $h_t^M, h_t^N$, respectively. Next, we apply MHA [24] separately to each set of context embeddings to obtain two intermediate outputs:

$$A^M = \mathrm{MHA}(q, k^M, v^M), \tag{13}$$

$$A^N = \mathrm{MHA}(q, k^N, v^N), \tag{14}$$

where $k^M$, $v^M$ and $k^N$, $v^N$ are the keys and values derived from $h_L^M$ and $h_L^N$, respectively. These outputs are linearly projected and aggregated:

$$A^2 = W_3^M A^M + W_3^N A^N, \tag{15}$$

where $W_3^M$ and $W_3^N$ are learnable matrices.

**Query Mapping and Feed-Forward Layer.** To further enrich the query representation, we introduce a query mapping transformation and a feed-forward layer with residual connection:

$$q' = A^2 + \mathrm{QMT}(q), \tag{16}$$

$$q^A = q' + \mathrm{FF}(q'), \tag{17}$$

where $\mathrm{QMT}(\cdot)$ is a linear projection that maps $q$ to the same dimensionality as $A^2$, defined as:

$$\mathrm{QMT}(q) = \delta^{\mathrm{QMT}} q. \tag{18}$$

Here, $\delta^{\mathrm{QMT}}$ is a learnable weight matrix. Given the final context vector $q^A$, we compute a score $\gamma_j$ for each node $j$ using a masked single-head attention mechanism:

$$\gamma_j = \begin{cases} C \cdot \tanh\left( \frac{q^A (k_j^M + k_j^N)}{\sqrt{d_k}} \right), & \text{if } j \text{ unvisited} \\ -\infty, & \text{otherwise} \end{cases} \tag{19}$$

where $d_k$ is the dimensionality of the key vectors, and $C$ is a scaling constant. The final selection probability for node $j$ is computed via the softmax function:

$$\rho_j = \mathrm{softmax}(\gamma_j). \tag{20}$$

At each decoding step, a node $\tau_j$ is sampled according to $\rho_j$. Repeating this process for $n$ steps yields the full solution $\tau = (\tau_1, ..., \tau_n)^T$. In addition, we report the training algorithm in Appendix B.

# 4 Experiments

We evaluate the performance of **UniteFormer** on synthetic TSP and CVRP instances of varying sizes, under three input settings: node-only, edge-only, and hybrid. We also report results on standard real-world benchmarks from TSPLib and CVRPLib. The code is publicly available[1].

---

[1] https://github.com/Regina921/UniteFormer

**Baselines.** 1) Traditional Solvers: Concorde [8], LKH3 [14], OR-Tools [25], and HGS [41]. 2) Learning-based Solvers: MatNet [23], GREAT [28], POMO [24], LEHD [31], GCN-BS [18], DAR [44] and ICAM [49]. In order to compare POMO with UniteFormer on edge-base input, we re-implement POMO using edge-only inputs (denoted as **POMO-edge**). More detailed descriptions of these baselines are presented in Appendix D.

**Problem Setting.** We follow the standard data generation procedures from prior work [24] to create training and testing datasets for TSP and CVRP with $n = 20, 50, 100$, where $n$ denotes the number of nodes. The problem setups and implementation details are presented in Appendix C.

**Model Setting.** The edge-aware sub-encoder consists of one layer of mixed-score MHA, three layers of GCN and MLP, and one self-attention layer. The node-focused sub-encoder consists of three attention layers. In each attention layer, the head number of MHA is set to 16, the embedding dimension is set to 256, and the feed-forward layer dimension is set to 512.

**Training and Inference.** We use the REINFORCE algorithm [46], training each model for 1,010 epochs with 100,000 instances per epoch. The Adam [20] optimizer is used with an initial learning rate of $4e^{-4}$ and weight decay is set to $1e^{-6}$. We adopt the POMO inference algorithm [24] and report both the optimality gap and inference time. A separate set of 10,000 uniformly generated instances is used for testing. All experiments were conducted on a single Tesla V100-SXM2-32GB GPU. More experiment setup details are presented in Appendix D.

## 4.1 Experimental Results

We train a single unified model capable of handling three input types: edge-only, node-only, and hybrid input. Table 1 reports the performance of **UniteFormer** on uniformly distributed TSP and CVRP instances across various problem sizes and input modalities. UniteFormer consistently outperforms existing learning-based methods in both greedy (×1) and instance-augmented (×8) inference, while maintaining competitive inference times. Additionally, following MatNet [23], we also report results under large-scale augmentation (×128) for edge-based input.

**TSP.** For edge-based input, **UniteFormer** significantly outperforms both POMO-edge, MatNet and GREAT across all sizes studied. Notably, its performance with ×8 augmentation exceeds that of MatNet's ×128 augmentation, highlighting the efficiency of the UniteFormer. For node-based input, UniteFormer achieves superior results over node-based neural solvers, including POMO and even the strong LEHD model, in both greedy and ×8 inference. For hybrid edge-node input, UniteFormer also surpasses methods such as GCN-BS, DAR, and ICAM across all scales studied. The advantage is particularly evident on TSP100, where UniteFormer achieves the lowest optimality gap of just 0.0589% among all neural baselines in Table 1. Furthermore, our edge-based UniteFormer even outperforms not only node-based models like POMO, but also hybrid models like DAR and ICAM, demonstrating its strong generalization and representational capacity.

**CVRP.** Similarly, for CVRP, **UniteFormer** exhibits robust performance across all input types. In the edge-based setting, UniteFormer outperforms both POMO-edge and MatNet in greedy and instance-augmented inference. Its performance with ×8 augmentation even exceeds MatNet's ×128 augmentation results. In the node-based and hybrid settings, UniteFormer again achieves the best results among all compared neural solvers. Specifically, on CVRP100, the hybrid-input version of UniteFormer achieves the lowest average optimality gap of 0.5963%. Notably, in the edge-only setting, UniteFormer even surpasses several node-based or hybrid methods, including POMO, DAR, and ICAM. These results comprehensively demonstrate the effectiveness, robustness, and versatility of UniteFormer across a range of problem sizes and input modalities.

## 4.2 Ablation Study

**Edge only vs. Node only vs. Edge and Node only vs. UniteFormer.** Table 2 presents the results of our ablation study comparing **UniteFormer** with three training variants. At test time, we evaluate models under three input configurations: *Input-edge* (only edge features are provided), *Input-node* (only node features are provided), and *Input-XE* (both edge and node features are available). The first variant, denoted as *w.o. UF-Edge*, is trained exclusively with edge inputs. The second variant,

Table 1: Experimental results on TSP and CVRP with uniformly distributed instances. The results of methods with an asterisk (#) are directly obtained from the original paper. BS: Beam search, BS*: Beam search and shortest tour heuristic. UniteFormer-E: Our UniteFormer takes edge as input; '-X': the UniteFormer takes node as input; '-XE': the UniteFormer takes edge and node as input.

| | Method | TSP20 | | | TSP50 | | | TSP100 | | |
| | | Len. | Gap(%) | Time(m) | Len. | Gap(%) | Time(m) | Len. | Gap(%) | Time(m) |
|---|---|---|---|---|---|---|---|---|---|---|
| | Concorde | 3.831 | 0.000 | 4.43 | 5.691 | 0.000 | 23.53 | 7.763 | 0.000 | 66.45 |
| | LKH3 | 3.831 | 0.000 | 2.78 | 5.691 | 0.000 | 17.21 | 7.763 | 0.000 | 49.56 |
| | OR-Tools | 3.864 | 0.864 | 1.16 | 5.851 | 2.795 | 10.75 | 8.057 | 3.782 | 39.05 |
| edge | POMO-edge | 3.837 | 0.164 | 0.10 | 5.719 | 0.482 | 0.24 | 7.919 | 2.003 | 1.34 |
| | MatNet(×1) | 3.832 | 0.044 | 0.11 | 5.709 | 0.303 | 0.13 | 7.836 | 0.940 | 0.52 |
| | MatNet(×8) | 3.831 | 0.002 | 0.22 | 5.694 | 0.050 | 1.24 | 7.795 | 0.410 | 5.28 |
| | MatNet(×128) | 3.831 | 0.000 | 5.71 | 5.692 | 0.013 | 16.47 | 7.776 | 0.170 | 60.11 |
| | GREAT(×1)# | - | - | - | - | - | - | 7.850 | 1.210 | 2.00 |
| | GREAT(×8)# | - | - | - | - | - | - | 7.820 | 0.810 | 18.00 |
| | UniteFormer-E(×1) | 3.831 | 0.020 | 0.11 | 5.697 | 0.095 | 0.32 | 7.789 | 0.332 | 1.61 |
| | UniteFormer-E(×8) | 3.831 | **0.000** | 0.22 | 5.692 | **0.004** | 1.05 | 7.770 | **0.086** | 5.12 |
| | UniteFormer-E(×128) | 3.831 | 0.000 | 2.74 | 5.691 | 0.000 | 19.67 | 7.765 | 0.019 | 65.21 |
| node | POMO(×1) | 3.831 | 0.018 | 0.08 | 5.698 | 0.119 | 0.24 | 7.792 | 0.364 | 1.03 |
| | POMO(×8) | 3.831 | 0.001 | 0.11 | 5.693 | 0.024 | 0.45 | 7.774 | 0.142 | 2.01 |
| | LEHD Greedy | 3.867 | 0.961 | 0.14 | 5.721 | 0.519 | 0.24 | 7.808 | 0.577 | 1.37 |
| | UniteFormer-X(×1) | 3.831 | 0.011 | 0.11 | 5.696 | 0.072 | 0.32 | 7.788 | 0.316 | 1.62 |
| | UniteFormer-X(×8) | 3.831 | 0.000 | 0.22 | 5.692 | 0.004 | 1.04 | 7.770 | 0.085 | 5.15 |
| edge+node | GCN | 3.855 | 0.650 | 0.25 | 5.901 | 3.678 | 1.21 | 8.413 | 8.373 | 6.25 |
| | GCN-BS | 3.835 | 0.128 | 0.81 | 5.710 | 0.317 | 4.62 | 7.931 | 2.155 | 17.73 |
| | GCN-BS* | 3.831 | 0.000 | 21.25 | 5.694 | 0.041 | 37.63 | 7.869 | 1.368 | 58.34 |
| | DAR(×1) | 3.831 | 0.021 | 0.08 | 5.702 | 0.181 | 0.26 | 7.803 | 0.512 | 1.12 |
| | DAR(×8) | 3.831 | 0.001 | 0.12 | 5.694 | 1.040 | 0.52 | 7.779 | 0.201 | 1.72 |
| | ICAM(×1) | 3.831 | 0.022 | 0.07 | 5.701 | 0.172 | 0.22 | 7.806 | 0.55 | 0.99 |
| | ICAM(×8) | 3.831 | 0.002 | 0.11 | 5.694 | 0.417 | 0.48 | 7.780 | 0.22 | 1.34 |
| | UniteFormer-XE(×1) | 3.831 | 0.009 | 0.11 | 5.695 | 0.055 | 0.32 | 7.782 | 0.243 | 1.62 |
| | UniteFormer-XE(×8) | 3.831 | 0.000 | 0.22 | 5.692 | 0.003 | 1.06 | 7.768 | 0.059 | 5.19 |

| | Method | CVRP20 | | | CVRP50 | | | CVRP100 | | |
| | | Len. | Gap(%) | Time(m) | Len. | Gap(%) | Time(m) | Len. | Gap(%) | Time(m) |
|---|---|---|---|---|---|---|---|---|---|---|
| | LKH3 | 6.117 | 0.000 | 2.15h | 10.347 | 0.000 | 8.52h | 15.647 | 0.000 | 13.46h |
| | HGS | 6.112 | -0.079 | 1.48h | 10.347 | -0.001 | 4.67h | 15.584 | -0.401 | 6.54h |
| | OR-Tools | 6.414 | 4.863 | 2.37 | 11.219 | 8.430 | 19.35 | 17.172 | 9.749 | 2.61h |
| edge | POMO-edge | 6.160 | 0.700 | 0.12 | 10.525 | 1.725 | 0.35 | 15.943 | 1.893 | 1.53 |
| | MatNet(×1) | 6.172 | 0.907 | 0.12 | 10.787 | 4.253 | 0.21 | 16.280 | 4.401 | 1.02 |
| | MatNet(×8) | 6.146 | 0.469 | 0.58 | 10.635 | 2.787 | 1.23 | 16.117 | 3.356 | 4.65 |
| | MatNet(×128) | 6.131 | 0.229 | 9.93 | 10.538 | 1.847 | 17.93 | 15.989 | 2.530 | 69.05 |
| | UniteFormer-E(×1) | 6.146 | 0.486 | 0.06 | 10.471 | 1.204 | 0.56 | 15.868 | 1.416 | 1.95 |
| | UniteFormer-E(×8) | 6.125 | **0.140** | 0.23 | 10.416 | **0.668** | 1.29 | 15.766 | **0.765** | 6.01 |
| | UniteFormer-E(×128) | 6.119 | 0.036 | 3.47 | 10.387 | 0.384 | 19.44 | 15.705 | 0.374 | 76.43 |
| node | POMO(×1) | 6.160 | 0.698 | 0.06 | 10.533 | 1.799 | 0.18 | 15.837 | 1.216 | 0.64 |
| | POMO(×8) | 6.132 | 0.254 | 0.20 | 10.437 | 0.875 | 0.48 | 15.754 | 0.689 | 2.11 |
| | LEHD Greedy | 6.462 | 5.647 | 0.07 | 10.872 | 5.075 | 0.18 | 16.217 | 3.648 | 0.55 |
| | UniteFormer-X(×1) | 6.145 | 0.454 | 0.06 | 10.472 | 1.210 | 0.56 | 15.856 | 1.335 | 1.95 |
| | UniteFormer-X(×8) | 6.126 | 0.155 | 0.22 | 10.419 | 0.694 | 1.26 | 15.753 | 0.672 | 6.05 |
| edge+node | DAR(×1) | 6.161 | 0.715 | 0.08 | 10.537 | 1.842 | 0.19 | 15.906 | 1.659 | 1.02 |
| | DAR(×8) | 6.132 | 0.240 | 0.13 | 10.441 | 0.907 | 0.49 | 15.783 | 0.873 | 2.23 |
| | ICAM(×1) | 6.160 | 0.703 | 0.08 | 10.502 | 1.504 | 0.13 | 15.955 | 1.972 | 0.62 |
| | ICAM(×8) | 6.132 | 0.246 | 0.12 | 10.439 | 0.886 | 0.39 | 15.833 | 1.192 | 1.77 |
| | UniteFormer-XE(×1) | 6.143 | 0.430 | 0.06 | 10.465 | 1.139 | 0.56 | 15.837 | 1.219 | 1.95 |
| | UniteFormer-XE(×8) | 6.126 | 0.146 | 0.22 | 10.415 | 0.660 | 1.28 | 15.740 | 0.596 | 6.05 |

*w.o. UF-Node*, is trained only with node inputs. The third variant, *w.o. UF-XE*, is trained solely on combined edge-node inputs. In contrast, our full **UniteFormer** model is trained using a hybrid strategy, where the input type (edge, node, or both) is randomly selected for each batch during training. As shown in Table 2, each variant performs well on its respective training input type but shows significant performance drops on other types. In contrast, UniteFormer consistently performs well in all input settings, demonstrating its ability to generalize effectively regardless of the input

Table 2: Ablations of three input variants of UniteFormer on uniformly distributed instances.

| TSP50 | | w.o. UF-Edge | | | w.o. UF-Node | | | w.o. UF-XE | | | UniteFormer | | |
|---|---|---|---|---|---|---|---|---|---|---|---|---|---|
| | | Len. | Gap(%) | Time(m) | Len. | Gap(%) | Time(m) | Len. | Gap(%) | Time(m) | Len. | Gap(%) | Time(m) |
| | Concorde | 5.691 | 0.000 | 23.53 | 5.691 | 0.000 | 23.53 | 5.691 | 0.000 | 23.53 | 5.691 | 0.000 | 23.53 |
| edge | input-edge(×1) | 5.694 | 0.041 | 0.32 | 11.922 | 109.479 | 0.32 | 12.349 | 116.976 | 0.32 | 5.697 | 0.095 | 0.32 |
| edge | input-edge(×8) | 5.692 | **0.002** | 1.05 | 10.033 | 76.278 | 1.05 | 10.161 | 78.524 | 1.04 | 5.692 | 0.004 | 1.05 |
| node | input-node(×1) | 8.424 | 48.010 | 0.32 | 5.698 | 0.121 | 0.32 | 5.951 | 4.566 | 0.32 | 5.696 | 0.072 | 0.32 |
| node | input-node(×8) | 7.473 | 31.293 | 1.04 | 5.692 | 0.011 | 1.05 | 5.801 | 1.929 | 1.05 | 5.692 | **0.004** | 1.04 |
| X+E | input-XE(×1) | 6.620 | 16.316 | 0.32 | 5.726 | 0.604 | 0.32 | 5.695 | 0.069 | 0.32 | 5.695 | 0.055 | 0.32 |
| X+E | input-XE(×8) | 6.225 | 9.370 | 1.04 | 5.696 | 0.085 | 1.04 | 5.692 | 0.003 | 1.04 | 5.692 | **0.003** | 1.04 |
| CVRP50 | | w.o. UF-Edge | | | w.o. UF-Node | | | w.o. UF-XE | | | UniteFormer | | |
| | | Len. | Gap(%) | Time(m) | Len. | Gap(%) | Time(m) | Len. | Gap(%) | Time(m) | Len. | Gap(%) | Time(m) |
| | LKH3 | 10.347 | 0.000 | 8.52h | 10.347 | 0.000 | 8.52h | 10.347 | 0.000 | 8.52h | 10.347 | 0.000 | 8.52h |
| edge | input-edge(×1) | 10.457 | 1.069 | 0.56 | 32.755 | 216.567 | 0.56 | 28.768 | 178.041 | 0.56 | 10.471 | 1.204 | 0.56 |
| edge | input-edge(×8) | 10.412 | **0.633** | 1.26 | 31.360 | 203.092 | 1.26 | 27.555 | 166.317 | 1.27 | 10.416 | 0.668 | 1.26 |
| node | input-node(×1) | 24.763 | 139.332 | 0.56 | 10.455 | 1.048 | 0.56 | 11.280 | 9.018 | 0.56 | 10.472 | 1.206 | 0.56 |
| node | input-node(×8) | 17.780 | 71.840 | 1.26 | 10.410 | **0.614** | 1.26 | 10.966 | 5.987 | 1.26 | 10.419 | 0.694 | 1.26 |
| X+E | input-XE(×1) | 12.491 | 20.720 | 0.56 | 10.455 | 1.048 | 0.56 | 10.449 | 0.991 | 0.56 | 10.465 | 1.139 | 0.56 |
| X+E | input-XE(×8) | 11.576 | 11.877 | 1.26 | 10.410 | 0.614 | 1.26 | 10.407 | **0.577** | 1.27 | 10.415 | 0.660 | 1.26 |

Table 3: Experimental results on TSPLIB and CVRPLIB.

| | Method | TSPLIB1-100 | | | TSPLIB101-300 | | | TSP301-500 | | |
|---|---|---|---|---|---|---|---|---|---|---|
| | | Len. | Gap(%) | Time(m) | Len. | Gap(%) | Time(m) | Len. | Gap(%) | Time(m) |
| | OPT | 19499.583 | 0.000 | - | 56902.800 | 0.000 | - | 35772.500 | 0.000 | - |
| edge | POMO-edge | 26434.493 | 38.177 | 0.11 | 86680.094 | 52.557 | 0.13 | 66931.096 | 85.899 | 0.24 |
| edge | MatNet(×1) | 20222.358 | 4.948 | 0.09 | 63182.822 | 9.061 | 0.19 | 50811.854 | 41.948 | 0.28 |
| edge | MatNet(×8) | 19748.707 | 2.119 | 0.16 | 61892.651 | 7.426 | 0.26 | 49611.249 | 38.678 | 0.37 |
| edge | UniteFormer-E(×1) | 20263.171 | 4.419 | 0.09 | 62497.070 | 11.322 | 0.12 | 49688.931 | 38.359 | 0.21 |
| edge | UniteFormer-E(×8) | 19723.757 | **1.621** | 0.17 | 58916.975 | **3.028** | 0.25 | 42414.251 | **18.308** | 0.35 |
| node | POMO(×1) | 19625.737 | 1.069 | 0.08 | 59779.103 | 3.405 | 0.11 | 48074.882 | 33.955 | 0.10 |
| node | POMO(×8) | 19552.937 | 0.781 | 0.14 | 59595.007 | 2.838 | 0.23 | 45171.931 | 26.200 | 0.11 |
| node | LEHD Greedy | 19891.001 | 2.382 | 0.11 | 58443.261 | 2.382 | 0.24 | 39391.629 | 10.630 | 0.32 |
| node | UniteFormer-X(×1) | 20079.525 | 4.316 | 0.09 | 58632.537 | 2.678 | 0.12 | 39599.271 | 10.807 | 0.22 |
| node | UniteFormer-X(×8) | 19549.664 | **0.614** | 0.18 | 57726.770 | **1.092** | 0.26 | 38983.657 | **9.252** | 0.35 |
| edge+node | DAR(×1) | 19767.780 | 1.981 | 0.08 | 58354.624 | 2.014 | 0.11 | 41488.967 | 15.856 | 0.21 |
| edge+node | DAR(×8) | 19616.985 | 1.123 | 0.14 | 57948.545 | 1.447 | 0.23 | 39834.717 | 12.376 | 0.33 |
| edge+node | ICAM(×1) | 19657.984 | 1.066 | 0.08 | 59456.423 | 3.125 | 0.11 | 41754.691 | 16.507 | 0.21 |
| edge+node | ICAM(×8) | 19622.840 | 0.843 | 0.14 | 58512.789 | 2.004 | 0.23 | 40771.396 | 13.923 | 0.31 |
| edge+node | UniteFormer-XE(×1) | 19916.707 | 2.585 | 0.10 | 58051.056 | 1.919 | 0.12 | 40213.122 | 12.482 | 0.21 |
| edge+node | UniteFormer-XE(×8) | 19567.880 | **0.653** | 0.18 | 57848.545 | **1.238** | 0.24 | 39775.667 | **11.111** | 0.35 |
| | Method | CVRPLIB1-100 | | | CVRP101-300 | | | CVRP301-500 | | |
| | | Len. | Gap(%) | Time(m) | Len. | Gap(%) | Time(m) | Len. | Gap(%) | Time(m) |
| | OPT | 915.574 | 0.000 | - | 33184.483 | 0.000 | - | 97160.000 | 0.000 | - |
| edge | POMO-edge | 1787.383 | 94.693 | 0.21 | 61150.353 | 108.337 | 0.23 | 195349.248 | 135.174 | 0.45 |
| edge | MatNet(×1) | 990.186 | 8.355 | 0.16 | 41257.808 | 27.962 | 0.25 | 119378.472 | 24.976 | 0.42 |
| edge | MatNet(×8) | 974.015 | 6.485 | 0.22 | 37060.391 | 12.766 | 0.32 | 111725.107 | 16.338 | 0.68 |
| edge | UniteFormer-E(×1) | 968.487 | 5.783 | 0.19 | 40601.853 | 23.299 | 0.25 | 112460.491 | 18.192 | 0.45 |
| edge | UniteFormer-E(×8) | 942.392 | **3.040** | 0.23 | 36259.083 | **9.485** | 0.38 | 110828.859 | **14.657** | 0.72 |
| node | POMO(×1) | 980.137 | 7.893 | 0.11 | 36514.578 | 9.260 | 0.18 | 113101.594 | 17.037 | 0.35 |
| node | POMO(×8) | 953.564 | 4.727 | 0.22 | 36004.930 | 7.561 | 0.26 | 110642.211 | 14.488 | 0.45 |
| node | LEHD Greedy | 1411.705 | 5.517 | 0.28 | 43841.532 | 11.734 | 0.45 | 116158.490 | 15.194 | 0.77 |
| node | UniteFormer-X(×1) | 953.203 | 4.266 | 0.19 | 35784.084 | 8.945 | 0.26 | 112799.943 | 13.897 | 0.45 |
| node | UniteFormer-X(×8) | 934.185 | **2.116** | 0.22 | 34971.075 | **5.353** | 0.38 | 105541.708 | **7.855** | 0.73 |
| edge+node | DAR(×1) | 959.278 | 5.060 | 0.18 | 35482.062 | 7.426 | 0.26 | 109347.025 | 9.449 | 0.44 |
| edge+node | DAR(×8) | 937.261 | 2.568 | 0.24 | 34990.418 | 5.530 | 0.37 | 104165.243 | 7.758 | 0.62 |
| edge+node | ICAM(×1) | 960.942 | 5.272 | 0.17 | 38232.511 | 11.489 | 0.25 | 117786.209 | 15.771 | 0.42 |
| edge+node | ICAM(×8) | 940.882 | 3.003 | 0.24 | 35881.780 | 7.338 | 0.39 | 110088.655 | 10.823 | 0.67 |
| edge+node | UniteFormer-XE(×1) | 951.386 | 3.994 | 0.19 | 36184.185 | 7.306 | 0.25 | 107177.981 | 8.798 | 0.45 |
| edge+node | UniteFormer-XE(×8) | 932.698 | **1.940** | 0.22 | 34989.507 | **5.146** | 0.38 | 104911.995 | **6.889** | 0.72 |

modality. This result highlights the strength of our unified training approach in producing a robust and versatile model.

Further architectural ablation studies of the three components of the UniteFormer architecture are reported in Appendix E. These experimental results also demonstrate their positive contribution to the UniteFormer, proving the effectiveness and indispensability of each component.

### 4.3 Generalization

Table 3 summarizes the results on real-world TSPLIB [36] and CVRPLIB [39] instances of various sizes and distributions. We categorize them into three groups by size: $N$=1-100, $N$=101-300, and $N$=301-500. Generalization results show that the UniteFormer performs best on instances with no more than 100 nodes and slightly less effectively on 101-300 node instances. Overall, UniteFormer shows excellent generalization ability on both TSPLIB and CVRPLIB. For TSP, UniteFormer generalizes better with node-only input. For CVRP, it performs exceptionally well with hybrid input.

Additionally, to demonstrate UniteFormer's strong generalization and scalability, we extend our investigation to the Asymmetric Traveling Salesman Problem (ATSP). Due to the asymmetric nature of ATSP, we can naturally solve it using the edge-based UniteFormer framework, which also exhibits superior performance. The detailed experimental results are shown in Appendix F.

## 5 Conclusion, Limitation and Future work

**Conclusion:** In this work, we propose UniteFormer, a unified neural solver that supports three input types through a single model trained via joint edge-node modalities. We propose a mixed encoder that integrates GCNs and attention mechanisms to collaboratively process node and edge features, capturing cross-modal interactions. Furthermore, we implement a parallel decoding strategy and enhance the decoder's representation ability by adding query mapping and nonlinear layers. Extensive experimental comparisons with other modality-specific models demonstrate UniteFormer's promising performance. Due to the efficiency and practicality of UniteFormer, we believe it can provide valuable insights and inspire follow-up work to explore more powerful unified neural solvers for edge-node modalities.

**Limitation and Future Work:** Although our UniteFormer performs well on all three input types, its heavy encoder results in high training and equipment demands, making large-scale problem training challenging. Trimming UniteFormer into a lightweight model for large-scale VRPs [48] is a worthwhile direction for future research. Another promising future work is to extend UniteFormer to solve multi-task VRPs [50, 5] with joint modalities.

## Acknowledgments

This work was supported in part by the National Natural Science Foundation of China under Grant 62372081, the Young Elite Scientists Sponsorship Program by CAST under Grant 2022QNRC001, the Liaoning Provincial Natural Science Foundation Program under Grant 2024010785-JH3/107, the Dalian Science and Technology Innovation Fund under Grant 2024JJ12GX020, the Dalian Major Projects of Basic Research under Grant 2023JJ11CG002 and the 111 Project under Grant D23006. This research/project is supported by the National Research Foundation, Singapore under its AI Singapore Programme (AISG Award No: AISG3-RP-2022-031).

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

# A    Details of Attention Computation

## A.1    Mixed-Score Attention

To incorporate external relational priors—such as graph adjacency or edge distances—alongside content-based similarity scores, we replace each head's standard scaled dot-product attention with a mixed-score attention mechanism, while preserving the remaining components of the Transformer's multi-head attention block (as in MatNet [23]). The original scaled dot-product attention computes attention weights by applying SoftMax to the pairwise scaled dot-products of queries and keys and then uses those weights to form a weighted sum of the value vectors. By contrast, the mixed-score attention supplements these internal attention scores with externally given relationship scores for every query-key pair. Specifically, the mixed-score attention mechanism mixes the internal attention scores and the external relationship scores before passing them to the next "SoftMax" stage. Similar mixing strategies have been explored by [38] and [11].

Formally, by defining dimensions $d_k$ and $d_v$, we compute the key $k_i \in \mathbb{R}^{d_k}$, value $v_i \in \mathbb{R}^{d_v}$, and query $q_i \in \mathbb{R}^{d_k}$ for each node by projecting the embedding $h_i$:

$$q_i = W^Q h_i, \quad k_i = W^K h_i, \quad v_i = W^V h_i, \tag{21}$$

where $W^Q \in \mathbb{R}^{d_k \times d_h}$, $W^K \in \mathbb{R}^{d_k \times d_h}$, and $W^V \in \mathbb{R}^{d_v \times d_h}$ are learnable weight matrices.

**Internal and External Scores:**    Afterwards, we compute the internal attention scores $S_{ij}^{\text{int}}$ by taking the scaled dot-products of each query–key pair and derive the external relationship scores $S_{ij}^{\text{ext}}$ from the edge weight matrix $D_{ij}$:

$$S_{ij}^{\text{int}} = \frac{q_i^T k_j}{\sqrt{d_k}}, \quad S_{ij}^{\text{int}} \in \mathbb{R}^{n \times n}, \tag{22}$$

$$S_{ij}^{\text{ext}} = g_\theta(D_{ij}), \quad S_{ij}^{\text{ext}} \in \mathbb{R}^{n \times n}, \tag{23}$$

where $S_{ij}^{\text{int}}$ denotes internal attention scores, $S_{ij}^{\text{ext}}$ denotes the external relationship scores, $D_{ij} \in \mathbb{R}^{n \times n}$ encodes a known relationship (edge weight matrix) between positions $(i, j)$, and $g_\theta$ is an optional learnable scalar or nonlinear mapping applied element-wise.

**Mixed Score via Element-Wise MLP:**    For each attention head $h$, we employ a compact two-layer perceptron $f_{\phi_h} : \mathbb{R}^2 \to \mathbb{R}$, parameterized by $\phi_h = \{W_1^h, b_1^h, W_2^h, b_2^h\}$, to fuse the internal and external scores on an element-wise basis:

$$S_{ij}^{\text{mix}} = f_{\phi_h}\left(S_{ij}^{\text{int}}, S_{ij}^{\text{ext}}\right) = W_2^h \, \sigma\left(W_1^h \, [S_{ij}^{\text{int}}, S_{ij}^{\text{ext}}]^\top + b_1^h\right) + b_2^h, \tag{24}$$

where $\sigma$ is the ReLU function, and $S_{ij}^{\text{mix}}$ denotes the mixed scores.

**Softmax Normalization:**    Then, we perform softmax normalization on the mixed scores $S_{ij}^{\text{mix}}$:

$$\alpha_{ij}^h = \frac{\exp\left(S_{ij}^{\text{mix}}\right)}{\sum_{j'=1}^n \exp\left(S_{ij'}^{\text{mix}}\right)}, \tag{25}$$

Subsequently, the attention vector $\alpha_{ij}^h$ is transformed into a convex combination of the messages $v_j^h$, and the specific-head attention output $z_i^h$ is obtained:

$$z_i^h = \sum_{j=1}^n \alpha_{ij}^h \, v_j^h, \quad \mathbf{Z}^h = [\, z_1^h, \ldots, z_n^h \,] \in \mathbb{R}^{d_h \times d_v}. \tag{26}$$

## A.2    Self-Attention

Self-attention is a mechanism that dynamically assigns weights to each position within a sequence, capturing global dependencies by allowing each element in the sequence to interact with itself and other elements. We interpret the attention mechanism in [40] as a weighted message-passing algorithm between nodes in a graph. The weight of the message *value* that a node receives from

its neighbors depends on the *compatibility* of its *query* with the *keys* of its neighbors. Formally, we define dimensions $d_k$ and $d_v$ and compute the key $k_i \in \mathbb{R}^{d_k}$, value $v_i \in \mathbb{R}^{d_v}$ and query $q_i \in \mathbb{R}^{d_k}$ for each node by projecting the embedding $h_i$:

$$q_i = W^Q h_i, \quad k_i = W^K h_i, \quad v_i = W^V h_i, \tag{27}$$

where $W^Q \in \mathbb{R}^{d_k \times d_h}$, $W^K \in \mathbb{R}^{d_k \times d_h}$, and $W^V \in \mathbb{R}^{d_v \times d_h}$ are learnable weight matrices. From the queries and keys, we compute the compatibility $u_{ij} \in \mathbb{R}$ of the query $q_i$ of node $i$ with the key $k_j$ of node $j$ as the (scaled) dot product:

$$u_{ij} = \begin{cases} \frac{q_i^T k_j}{\sqrt{d_k}}, & \text{if } i \text{ adjacent to } j \\ -\infty. & \text{otherwise} \end{cases} \tag{28}$$

From the compatibilities $u_{ij}$, we compute the *attention weights* $a_{ij} \in [0, 1]$ using a softmax function:

$$a_{ij} = \frac{e^{u_{ij}}}{\sum_{j'} e^{u_{ij'}}}. \tag{29}$$

After that, the vector $h_i'$ that is received by node $i$ is the convex combination of messages $v_j$:

$$h_i' = \sum_j a_{ij} v_j. \tag{30}$$

Finally, we normalize the vector $h_i$ and use the feed-forward sublayer to compute the node-by-node projection:

$$\hat{h}_i' = \text{NORM}\left(h_i + h_i'\right), \tag{31}$$

$$h_i^c = \text{NORM}\left(\hat{h}_i' + \text{FF}\left(\hat{h}_i'\right)\right), \tag{32}$$

where, FF$(\cdot)$ is a feed-forward layer, and NORM$(\cdot)$ applies batch normalization. The feed-forward sublayer computes node-wise projections using a hidden (sub)sublayer with dimension $d_{\text{ff}} = 512$ and a ReLU activation. We use batch normalization with learnable $d_h$-dimensional affine parameters $w^{\text{bn}}$ and $b^{\text{bn}}$. The two sublayers are defined in detail as follows:

$$\text{FF}(\hat{h}_i) = W^{\text{ff},1} \cdot \text{ReLU}(W^{\text{ff},0} \hat{h}_i + b^{\text{ff},0}) + b^{\text{ff},1}, \tag{33}$$

$$\text{BN}(h_i) = w^{\text{bn}} \odot \overline{\text{BN}}(h_i) + b^{\text{bn}}. \tag{34}$$

Here, $\odot$ denotes the element-wise product and $\overline{\text{BN}}$ refers to batch normalization without affine transformation.

## B  Training Details

We adopt the same reinforcement learning framework as POMO [24], using the REINFORCE algorithm [46] to train the UniteFormer. At each training step, we sample a set of $n$ solution trajectories $\{\tau^1, \cdots, \tau^n\}$, compute their corresponding rewards $f(\tau^i)$, and apply approximate gradient ascent to maximize the expected return $\mathcal{L}$. The gradient of the objective $\mathcal{L}(\theta)$ with respect to model parameters $\theta$ is estimated as:

$$\nabla_\theta \mathcal{L}(\theta) \approx \frac{1}{n} \sum_{i=1}^n [(f(\tau^i) - b^i(s)) \nabla \log p_\theta(\tau^i | s)], \tag{35}$$

where $b^i(s)$ is a baseline used to reduce variance in the gradient estimate. Following common practice, we use a shared baseline defined as the average reward across the sampled trajectories:

$$b^i(s) = b_{\text{shared}}(s) = \frac{1}{n} \sum_{i=1}^n f(\tau^i), \tag{36}$$

and the probability of a trajectory $\tau^i$ under the policy is factorized as:

$$p_\theta(\tau^i \mid s) = \prod_{t=2}^M p_\theta(a_t^i \mid s, a_{1:t-1}^i), \tag{37}$$

where $a_t^i$ denotes the action at step $t$ in trajectory $\tau^i$, and $M$ is the length of the solution.

Table 4: Experiment Hyperparameters.

| Hyperparameter | Value | Hyperparameter | Value |
|---|---|---|---|
| **Model** | | **Training** | |
| Embedding dimension $d_h$ | 256 | Input choice $\psi_c$ | edge/node/edge+node |
| Number of attention heads $M_h$ | 16 | Batch size | 1024/256/64 |
| Number of encoder layers $L_e$ | 3 | Optimizer | Adam |
| Number of GCN layers $L_g$ | 3 | Learning rate (LR) | $4e^{-4}$ |
| Number of MLP layers $L_m$ | 3 | Weight decay | $1e^{-6}$ |
| K-nearest neighbors | 20 | LR scheduler | MultiStepLR |
| Feedforward hidden dimension $d_f$ | 512 | LR milestones | [901,1001] |
| Feedforward activation | ReLU | LR gamma | 0.1 |
| Tanh clipping $\xi$ | 10.0 | Train data per epoch | 100,000 |
| Normalization | Batch | Training epochs | 1010 |

## C  Implementation Details for TSP and CVRP

### C.1  Problem Setup

**TSP.** Solving a TSP instance with $n$ nodes requires finding the shortest loop that visits each node exactly once and eventually returns to the first visited node, where the distance between two nodes is the Euclidean distance. We generate TSP instances following AM [22], where the coordinates of $n$ nodes are randomly and uniformly sampled from the unit square.

**CVRP.** The CVRP instance involves $n$ customer nodes and one depot node, where the coordinates of the customer nodes and the depot node are uniformly sampled from the unit square. Each customer node $i$ has a normalized demand $\hat{\delta}_i = \delta_i/D$, where $\delta_i$ is sampled from the discrete set $\{1, 2, ..., 9\}$ and the vehicle capacity $D = 30, 40, 50$ for problem sizes $N = 20, 50, 100$, respectively. A delivery vehicle with unit capacity makes round trips starting and ending at the depot, delivering goods to customer nodes according to their demands and replenishing inventory at the depot, where each customer node is only allowed to be visited once. Our objective is to determine the shortest feasible set of routes that visits all nodes while respecting the vehicle's capacity constraint $D$.

### C.2  Implementation Details

For a TSP/CVRP instance $G = \{X, E\}$, the node features $\{x_1, ..., x_n\}$ are the $2D$-coordinates of the $n$ nodes in the graph, and the edge features $e(x_i, x_j) \in E$ are the edges between nodes $x_i$ and $x_j$ in the graph $G$. Our UniteFormer supports three input modalities: 1) *Edge-only input*, where only edge weights $E$ are provided; 2) *Node-only input*, where only node coordinates $X$ are provided; 3) *Hybrid input*, where both node coordinates $X$ and edge weights $E$ are available.

To formalize this, we introduce a modality-selection function $\psi_c \in \{0, 1, 2\}$, where

$$\psi_c = \begin{cases} 0, & \text{edge-only input,} \\ 1, & \text{node-only input,} \\ 2, & \text{hybrid input.} \end{cases} \tag{38}$$

During training, we employ the REINFORCE algorithm to randomly select $\psi_c$ at each batch. This stochastic modality sampling encourages the model to learn robust representations under all three input scenarios.

## D  Experimental Details for UniteFormer

### D.1  Experiment Baselines

**1) Traditional Solvers:** For TSP, we use the non-learning solvers Concorde [8], LKH3 [14], and OR-Tools [25], which are known for providing strong results on TSP. Consistent with prior works [24,

Table 5: Ablations of three key components of UniteFormer on uniformly distributed instances.

| | TSP50 | w.o. self-attention | | | w.o. node-focused sub | | | w.o. edge-aware sub-E | | | UniteFormer | | |
|---|---|---|---|---|---|---|---|---|---|---|---|---|---|
| | | Len. | Gap(%) | Time(m) | Len. | Gap(%) | Time(m) | Len. | Gap(%) | Time(m) | Len. | Gap(%) | Time(m) |
| | Concorde | 5.691 | 0.000 | 23.53 | 5.691 | 0.000 | 23.53 | 5.691 | 0.000 | 23.53 | 5.691 | 0.000 | 23.53 |
| edge | input-edge(×1) | 5.699 | 0.126 | 0.32 | 5.697 | 0.097 | 0.34 | 6.402 | 12.490 | 0.32 | 5.697 | 0.095 | 0.32 |
| | input-edge(×8) | 5.692 | 0.007 | 1.01 | 5.692 | 0.004 | 1.15 | 5.765 | 1.294 | 1.06 | 5.692 | **0.004** | 1.05 |
| node | input-node(×1) | 5.697 | 0.098 | 0.31 | 5.695 | 0.069 | 0.34 | 5.6978 | 0.111 | 0.33 | 5.696 | 0.072 | 0.32 |
| | input-node(×8) | 5.692 | 0.009 | 1.02 | 5.692 | 0.004 | 1.14 | 5.692 | 0.006 | 1.06 | 5.692 | **0.004** | 1.04 |
| X+E | input-XE(×1) | 5.695 | 0.068 | 0.31 | 5.695 | 0.054 | 0.35 | 5.697 | 0.105 | 0.35 | 5.695 | 0.055 | 0.32 |
| | input-XE(×8) | 5.692 | 0.004 | 1.02 | 5.692 | 0.003 | 1.16 | 5.692 | 0.005 | 1.08 | 5.692 | **0.003** | 1.04 |

| | CVRP50 | w.o. self-attention | | | w.o. node-focused sub | | | w.o. edge-aware sub-E | | | UniteFormer | | |
|---|---|---|---|---|---|---|---|---|---|---|---|---|---|
| | | Len. | Gap(%) | Time(m) | Len. | Gap(%) | Time(m) | Len. | Gap(%) | Time(m) | Len. | Gap(%) | Time(m) |
| | LKH3 | 10.347 | 0.000 | 8.52h | 10.347 | 0.000 | 8.52h | 10.347 | 0.000 | 8.52h | 10.347 | 0.000 | 8.52h |
| edge | input-edge(×1) | 10.483 | 1.312 | 0.54 | 10.529 | 1.763 | 0.58 | 15.111 | 46.043 | 0.56 | 10.471 | 1.204 | 0.56 |
| | input-edge(×8) | 10.426 | 0.770 | 1.24 | 10.434 | 0.841 | 1.32 | 14.203 | 37.266 | 1.29 | 10.416 | **0.668** | 1.26 |
| node | input-node(×1) | 10.490 | 1.379 | 0.54 | 10.544 | 1.909 | 0.58 | 10.677 | 3.192 | 0.56 | 10.472 | 1.206 | 0.56 |
| | input-node(×8) | 10.421 | 0.721 | 1.24 | 10.461 | 1.103 | 1.26 | 10.537 | 1.835 | 1.28 | 10.419 | **0.694** | 1.26 |
| X+E | input-XE(×1) | 10.470 | 1.192 | 0.56 | 10.531 | 1.781 | 0.59 | 10.660 | 3.028 | 0.56 | 10.465 | 1.139 | 0.56 |
| | input-XE(×8) | 10.417 | 0.680 | 1.25 | 10.453 | 1.028 | 1.35 | 10.526 | 1.734 | 1.28 | 10.415 | **0.660** | 1.26 |

31], we calculate the performance gap relative to Concorde. For CVRP, we use the non-learning solvers LKH3 [14], HGS [41], and OR-Tools [25]. We calculate the performance gap relative to LKH3.

**2) Learning-based Solvers:** *Edge-only input:* We compare with the current state-of-the-art Mat-Net [23] and GREAT [28], both of which use edge relationships as input for the problems. Among them, MatNet is retrained and evaluated on symmetric instances according to the original model design. Since GREAT has not published the code and data, the results are taken from their paper. Additionally, to fairly compare POMO-based models with UniteFormer, we also re-implemented POMO using edge-only input (denoted as *POMO-edge*). **Node-only input:** We use the public models of the classic node-based POMO [24] and LEHD [31] for comparative testing. **Hybrid input:** GCN-BS [18], DAR [44] and ICAM [49] all use edge and node as model input. Since the codes of DAR and ICAM are unavailable, we re-implement and retrain DAR and ICAM according to the official settings, and the number of training epochs and data sets are consistent with our UniteFormer.

### D.2 Experimental Hyperparameters

We report the hyperparameter details common across the main experiments in Table 4. In the table, "Input choice" indicates that during training we randomly select one of the three input modalities for each batch. We employ three layers for the GCN, the MLP, and the node-focused sub-encoder. All normalization operations within the model are implemented using batch normalization [16].

## E   Ablation Study of UniteFormer Architecture

Table 5 reports ablation studies comparing UniteFormer against three structural variants to isolate the contributions of each component. One may ask: why incorporate self-attention into the edge-aware sub-encoder? Why disable the node-focused sub-encoder when edge-only input is used? And what if the edge-aware sub-encoder were to operate solely on edge features, ignoring node attributes?

To answer these questions, we design three ablations: The first variant removes self-attention from the edge-aware sub-encoder (denoted by **w.o. self-attention**). The second variant adds the node-focused sub-encoder, transferring temporary node features to it when edge is as input (denoted by **w.o. node-focused sub**). The third variant feeds only edge features into the edge-aware sub-encoder, setting node features to zero embeddings (denoted as **w.o. edge-aware sub-E**). Across all input modalities, each variant trails the full UniteFormer configuration, as shown in Table 5. These results confirm that (a) self-attention in the edge-aware encoder is crucial for capturing edge-node interactions, (b) adding the node-focused sub-branch under edge-only input degrades performance, and (c) jointly leveraging both node and edge features yields superior representations. Overall, the ablation study demonstrates each component's positive contribution to the model and validates the effectiveness and design rationality of the UniteFormer architecture.

Table 6: Experimental results on 10,000 instances of ATSP.

| Method | ATSP20 | | | ATSP50 | | | ATSP100 | | |
|---|---|---|---|---|---|---|---|---|---|
| | Len. | Gap(%) | Time($m$) | Len. | Gap(%) | Time($m$) | Len. | Gap(%) | Time($m$) |
| CPLEX | 1.540 | 0.000 | 12.14 | 1.559 | 0.000 | 60.03 | 1.571 | 0.000 | 5.01h |
| Nearest Neighbor | 2.010 | 30.390 | - | 2.101 | 34.610 | - | 2.140 | 36.100 | - |
| Nearest Insertion | 1.800 | 16.560 | - | 1.950 | 25.160 | - | 2.050 | 30.790 | - |
| Furthest Insertion | 1.710 | 11.230 | - | 1.840 | 18.220 | - | 1.940 | 23.370 | - |
| LKH3 | 1.540 | 0.000 | 0.12 | 1.560 | 0.000 | 0.24 | 1.570 | 0.000 | 1.12 |
| MatNet($\times$1) | 1.548 | 0.533 | 0.06 | 1.580 | 1.350 | 0.15 | 1.622 | 3.242 | 0.62 |
| MatNet($\times$8) | 1.542 | 0.084 | 0.31 | 1.566 | 0.472 | 1.23 | 1.603 | 2.086 | 4.24 |
| MatNet($\times$128) | 1.540 | 0.012 | 4.72 | 1.561 | 0.144 | 16.55 | 1.590 | 0.934 | 61.03 |
| UniteFormer($\times$1) | 1.543 | 0.245 | 0.15 | 1.573 | 0.917 | 0.24 | 1.621 | 3.170 | 0.76 |
| UniteFormer($\times$8) | 1.540 | 0.043 | 0.36 | 1.564 | 0.339 | 1.22 | 1.600 | 1.841 | 5.07 |
| UniteFormer($\times$128) | 1.540 | **0.001** | 4.01 | 1.561 | **0.129** | 18.09 | 1.585 | **0.880** | 1.32h |

# F  Asymmetric Traveling Salesman Problem (ATSP)

## F.1  Problem Setup

In the classic TSP, the objective is to determine a tour over $N$ nodes that minimizes the total round-trip distance. For any two nodes $x_i$ and $x_j$, the edge distance satisfies $d(x_i, x_j) = d(x_j, x_i)$, yielding an $N \times N$ symmetric distance matrix. To demonstrate UniteFormer's strong generalization and scalability beyond this symmetric setting, we extend our investigation to the Asymmetric Traveling Salesman Problem (ATSP). In ATSP, one seeks the shortest Hamiltonian circuit in a directed, weighted graph that visits each vertex exactly once and returns to the start. Unlike the classic TSP, the ATSP's distance matrix is non-symmetric, with edge weights satisfying $d(x_i, x_j) \neq (x_j, x_i)$. Thus, it must accommodate both directional and weight asymmetries.

Our edge-only UniteFormer framework naturally accommodates these asymmetric instances without modification and continues to exhibit excellent performance. Following MatNet's experimental protocol, we evaluate on tmat class ATSP instances that satisfy the triangle inequality and are widely used in the operations-research (OR) community [7]. We solve three problem sizes ($N$=20, 50, and 100) and confirm that UniteFormer retains its robustness and scalability under the more general ATSP setting.

## F.2  Experiment Results

Our UniteFormer is able to take edge-only information as input and can be easily extended to solve ATSP. We use randomly generated asymmetric distance matrices as input. Both the training and test datasets follow the data generation method of MatNet.

Table 6 reports the performance of our trained model compared with other representative baseline algorithms on 10,000 test instances of tmat class ATSP. For each method, we report the average tour length (shown in units of $10^6$) and the percentage gap relative to CPLEX's optimal solutions. One of the key baselines compared in the table is the MatNet, which is an edge-based construction method designed for solving ATSP.

The results in Table 6 demonstrate that UniteFormer consistently outperforms MatNet under both greedy inference ($\times$1) and instance augmentation ($\times$8, $\times$128) inference, while maintaining relatively reasonable inference times. In particular, on the largest ATSP100 instances, our UniteFormer attains a modest performance advantage, albeit with a slightly increased runtime. These experiments confirm that our edge-driven UniteFormer achieves competitive performance on ATSP, showcasing its extensive adaptability and strong generalization. Collectively, they validate the effectiveness of the UniteFormer architecture for a broad spectrum of optimization challenges.

# G  Broader impacts

The paper introduces UniteFormer, a unified neural solver for VRPs that flexibly accommodates three distinct input modalities through joint edge–node training. This design significantly enhances the practicality of neural solvers for real-world applications. Moreover, the underlying techniques generalize beyond VRPs, opening the door to a wide array of combinatorial optimization tasks.

Our experiments demonstrate that, compared to state-of-the-art neural solvers trained on a single input modality, UniteFormer consistently delivers superior performance across diverse VRP variants. This unified approach not only streamlines training by eliminating the need for separate models per modality but also reduces overall engineering complexity. Nonetheless, the heavy encoder underpinning UniteFormer demands substantial computational resources, which poses a challenge for scaling to very large problem instances. Future work will focus on architecting more lightweight yet expressive models to bridge this gap.

