# OpenReview forum: "UniteFormer: Unifying Node and Edge Modalities in Transformers for Vehicle Routing Problems"
_NeurIPS.cc/2025/Conference — NeurIPS 2025 spotlight_

### Official Review · Reviewer_HjGY · 2025-06-30

**Clarity:** 3
**Significance:** 3
**Originality:** 3
**Rating:** 5
**Confidence:** 4

**Summary:**

This paper introduces UniteFormer, a novel neural solver for Vehicle Routing Problems (VRPs) designed to overcome the limitations of existing methods that typically rely on either node-only or edge-only inputs. UniteFormer proposes a unified framework that can handle node-only, edge-only, and hybrid (node and edge) input types using a single trained model, eliminating the need for retraining for different input modalities.

The core contributions of UniteFormer are:

A novel mixed encoder that integrates residual gated Graph Convolutional Networks (GCNs) with attention mechanisms. This encoder consists of an edge-aware sub-encoder and a node-focused sub-encoder, collaboratively processing both node and edge features to capture cross-modal interactions.

A parallel-attention decoder enhanced with query mapping and a feed-forward layer to improve its representational capacity.
The model is trained using REINFORCE by randomly sampling input types across batches, allowing it to generalize across modalities.

Extensive experiments on TSP and CVRP, including real-world TSPLib and CVRPLib instances, demonstrate that UniteFormer achieves state-of-the-art performance across all three input types and shows strong generalization capabilities.

**Questions:**

Question: Given that REINFORCE can suffer from high variance, what specific techniques (e.g., baseline choices, larger batch sizes, other variance reduction methods) were employed to ensure stable and efficient training of UniteFormer? Were other policy gradient algorithms (like Actor-Critic variants or PPO) considered, and if so, what were the reasons for choosing REINFORCE over them?

**Ethical Concerns:**

["NO or VERY MINOR ethics concerns only"]

**Limitations:**

Yes.

**Paper Formatting Concerns:**

No.

**Quality:**

3

**Strengths And Weaknesses:**

Strengths:

Quality: UniteFormer's architecture is robust and well-designed, effectively addressing the challenge of unifying node and edge modalities. The mixed encoder with its two sub-encoders (edge-aware and node-focused) and the parallel-attention decoder show a sophisticated approach to integrating diverse information. The experimental results are comprehensive, covering TSP and CVRP on synthetic and real-world benchmarks (TSPLib, CVRPLib) across all three input settings (node-only, edge-only, hybrid), demonstrating state-of-the-art performance. The authors also re-implemented POMO for edge-only input for a fair comparison, which adds to the quality of the experimental setup.

Clarity: The paper is clearly written and well-structured. The problem statement regarding single-modality limitations is articulated effectively. The architecture of UniteFormer is explained logically, with a helpful overview diagram (Figure 1) and detailed breakdowns of the input modalities , sub-encoders , and decoder. Mathematical formulations are provided for key components, enhancing understanding.

Significance: The primary significance lies in proposing the first unified neural solver for VRPs that inherently supports node-only, edge-only, and hybrid inputs within a single trained model. This addresses a major practical limitation in current neural VRP solvers, offering greater flexibility and applicability to diverse real-world scenarios where input data might vary. The improved generalization and state-of-the-art performance contribute to advancing the field of neural combinatorial optimization for VRPs.

Originality: The concept of a single model capable of handling multiple input modalities (node, edge, hybrid) without retraining is highly original in the context of neural VRP solvers. The specific architectural designs, particularly the mixed encoder with its dual sub-encoders and their collaborative processing of heterogeneous features, represent a novel approach to information fusion. The random sampling of input types during REINFORCE training is also an original aspect of their methodology for achieving modality generalization.

Weaknesses:

Training Complexity and Hyper-parameter Sensitivity: The paper utilizes the REINFORCE algorithm for training, which is known for its high variance and potential training instability. However, the paper does not provide detailed insights into how these challenges were managed (e.g., specific variance reduction techniques, choice of baselines, or batch size considerations). While performance is good, a discussion on the robustness of the training process would be beneficial.

---

> ### Author Rebuttal · Authors · 2025-07-31
>
> Thank you for your constructive and valuable comments and suggestions. We address your concerns point-by-point as follows:
>
> >**Q1: Given that REINFORCE can suffer from high variance, what specific techniques (e.g., baseline choices, larger batch sizes, other variance reduction methods) were employed to ensure stable and efficient training of UniteFormer?**
>
> **A1:** Thank you for your question regarding the stability of REINFORCE training in UniteFormer. To mitigate the known high variance issue of REINFORCE, we adopt a shared baseline strategy, as detailed in **Appendix B**, which is also widely used and validated in prior works such as POMO and MatNet.
>
> **(1) A shared baseline strategy:** Specifically, we sample multiple trajectories per instance (e.g., via augmentation or diverse decoding), and use the average reward across these trajectories as the baseline. This shared baseline significantly reduces gradient variance and has been shown to improve training stability in VRP tasks. Empirical evidence from POMO shows that this shared baseline is more robust to local minima than the greedy rollout baseline in AM. This technique ensures that the advantage term $R(\tau_i)-b(s)$ has zero mean, and detailed formulas for the gradient variance can be found in **Appendix B**.
>
> In addition, we have adopted Batched Trajectory Sampling and Large Mini-Batch Training techniques to ensure more stable and efficient training of UniteFormer.
>
> **(2) Batched Trajectory Sampling:** Similar to POMO, we employ parallel exploration from diverse starting states, which leads to multiple diverse yet optimal solution representations. This practice effectively acts as a kind of entropy-regularized exploration, and avoids overfitting to narrow trajectory spaces, i.e., a common cause of instability in REINFORCE. Moreover, by leveraging the symmetries in routing problems (e.g., permutation invariance in TSP tours), we encourage the policy to generalize more robustly, thereby improving sample efficiency.
>
> **(3) Large Mini-Batch Training:** As suggested in prior literature (e.g., AM and POMO), we train the policy network with larger mini-batches (e.g., 64–256 problem instances per batch), which smooths out noisy gradients and contributes to more stable convergence.
>
> In summary, we combine a low-variance shared baseline, multiple trajectory rollouts, and large batch training to ensure that the REINFORCE training of UniteFormer remains stable and sample-efficient. These strategies have been shown to be effective in prior work (especially POMO) and are further validated in our empirical results across TSP, CVRP, and ATSP tasks.
>
> We appreciate your feedback again and will include a more explicit discussion in Appendix B.
>
>
> > **Q2: Were other policy gradient algorithms (like Actor-Critic variants or PPO) considered, and if so, what were the reasons for choosing REINFORCE over them?**
>
> **A2:** Thank you for your thoughtful question. In our work, we adopt the REINFORCE algorithm as the underlying policy gradient method, following a widely recognized and empirically validated precedent in neural combinatorial optimization, especially for routing problems. This choice is grounded in both empirical evidence and practical considerations, as discussed and validated in prior influential works such as AM and POMO. Particularly,
>
> * AM explicitly states in the abstract:
>
>   > *"We show how to train this model using REINFORCE with a simple baseline based on a deterministic greedy rollout, which we find is more efficient than using a value function."*
>
> * Similarly, POMO adopts REINFORCE and addresses variance through a shared baseline:
>
>   > *"We optimize the model using the REINFORCE algorithm, with a shared baseline across the multiple trajectories per instance. This baseline significantly reduces the variance of the gradient and improves stability."*
>
> These works demonstrate that, when paired with appropriate variance reduction techniques (e.g., rollout or shared baselines), REINFORCE can train powerful and generalizable neural solvers. As a result, REINFORCE has become a de facto standard in the VRP literature.
>
> In addition, we chose REINFORCE over more complex policy gradient algorithms such as Actor-Critic variants or PPO for several reasons:
>
> **(1) Simplicity and robustness in discrete, structured action spaces:** Routing problems involve large, dynamic, and discrete action spaces (e.g., permutations). Designing a stable critic network that generalizes over such combinatorial structures is highly non-trivial and often leads to convergence issues.
>
> **(2) Compatibility with autoregressive decoding:** Our decoder constructs solutions sequentially over permutations. REINFORCE allows us to compute unbiased policy gradients directly from sampled trajectories, without the need for approximating value functions over structured outputs.
>
> **(3) Lack of evidence for superior performance of Actor-Critic methods:** To our knowledge, actor-critic or PPO-style algorithms have not shown consistent advantages over REINFORCE in VRP tasks. In fact, empirical attempts to integrate these methods (e.g., applying PPO to POMO according to our preliminary experiments) have often resulted in non-convergent training. This further explains why recent state-of-the-art models continue to rely on REINFORCE-based frameworks.
>
> As detailed in **Appendix B**, our training setup with REINFORCE, which is combined with shared baselines, large batches, and architecture design, results in stable convergence and high-quality solutions across TSP, CVRP, and ATSP benchmarks.
>
> We acknowledge that more advanced policy gradient algorithms could be interesting for future work, especially if tailored to handle structured action spaces with learned critics. However, for the scope of this paper, REINFORCE strikes the best balance between effectiveness, reproducibility, and compatibility with our architecture. So we find REINFORCE to be the most suitable and efficient choice for our setting.

---

### Official Review · Reviewer_Fawv · 2025-07-01

**Clarity:** 3
**Significance:** 2
**Originality:** 2
**Rating:** 5
**Confidence:** 2

**Summary:**

This paper introduces UniteFormer, a Transformer-based method that processes nodes and edges simultaneously to solve vehicle routing problems (VRPs). The authors present a network architecture that can handle multimodal inputs. This architecture enables edge and node features to interact while maintaining the independence of their respective subnetworks.

**Questions:**

- Are existing methods inadequate simply because they use masked inputs for training? Graph neural networks are generally assumed to learn to complement node and edge features internally.
- If the network architecture is novel, then it should not be confined to VRPs. Could it benefit other tasks? Alternatively, is there a specific argument that handling edge-, node-, and hybrid inputs simultaneously is uniquely important for VRPs?
- The order and interplay of GCNs and attention are nontrivial. If attention alone is expected to yield rich, modality-independent features, then the absence of a GCN seems odd. Can you provide evidence supporting the soundness of this architectural choice?

**Ethical Concerns:**

["NO or VERY MINOR ethics concerns only"]

**Final Justification:**

Taking into account the responses to my concerns, as well as the discussions and evaluations provided by other reviewers, I agree with the positive overall sentiment and am inclined to support acceptance.

**Limitations:**

yes

**Paper Formatting Concerns:**

No concerns.

**Quality:**

2

**Strengths And Weaknesses:**

### Strengths
The loosely coupled design enables a single model to uniformly handle edge-, node-, and hybrid-only inputs. The method is evaluated in a reinforcement-learning setting for VRPs and demonstrates strong performance.

### Weaknesses
The review of existing work is insufficient, which leaves the novelty of the method unclear. Similar architectures to the proposed one are well known in graph-based neural networks, including works such as [*1] and [*2]. Comparable ideas have also been introduced in diverse fields, such as graph anomaly detection [*3], bipartite graph matching [*4], point-cloud registration [*5], and molecular modeling [*6]. Therefore, a broader survey with appropriate citations is desirable.

Such a survey would clarify the paper's contribution. It would demonstrate that a network capable of handling edges and vertices can accommodate edge-only, vertex-only, or hybrid inputs, particularly for VRPs. However, this is mainly a case study, so the degree of novelty is limited.

### Comment
Please change "Related Works" to "Related Work."

- [*1] Z. Hu+. Heterogeneous Graph Transformer. WWW2020.
- [*2] Z. Gao. A Graph is Worth K Words: Euclideanizing Graph using Pure Transformer. ICML2024.
- [*3] S. Yang. AHEAD: A Triple Attention Based Heterogeneous Graph Anomaly Detection Approach
- [*4] S. Sone+. WeaveNet for Approximating Two-sided Matching Problems. arXiv:2310.12515.
- [*5] R. Yanagi+. Learning 3D Point Cloud Registration as a Single Optimization Problem. ACCV2024.
- [*6] M. Anselmi+, Molecular graph transformer: stepping beyond ALIGNN into long-range interactions. Digital Discovery, 2024.

---

> ### Author Rebuttal · Authors · 2025-07-31
>
> Thank you for your constructive and valuable comments and suggestions.
> > **Q0:Related Work**
>
> **A0: (1) Scope of prior work:** Our study builds on an established body of research in neural VRP solvers. Prior works such as POMO, LEHD, RELD, GCN-BS, and Pointerformer (most of them are published in NeurIPS/ICLR) predominantly adopt GNNs or Transformer-based architectures within the VRP domain. To remain focused and highlight our specific contribution to modality-flexible neural solvers, we limited our related work to this area. Our motivation further stems from the lack of models that effectively integrate GNNs and Transformers for VRPs under varying or incomplete input modalities, which guided the design of our unified architecture.
>
> **(2) Broader works outside VRPs:** We appreciate the reviewer highlighting six relevant works[*1]–[*6] from other domains. While they share certain architectural elements, our method differs fundamentally in both objective and design, e.g.,: HGT[*1] models type-specific message passing in heterogeneous graphs, whereas UniteFormer addresses modality heterogeneity (node/edge/hybrid) for route optimization. GraphGPT[*2] focuses on Euclideanizing graphs for representation learning; we focus on path construction by jointly encoding structure and distances. AHEAD[*3] detects graph anomalies via triple-attention, while UniteFormer uses REINFORCE-based training to generate feasible tours in VRPs. WeaveNet[*4] solves bipartite matching over complete graphs; UniteFormer supports incomplete or partial input modalities, which is critical for real-world VRPs. [*5] addresses 3D point cloud registration using geometric alignment; our work is focused on graph-based sequential decision-making. MGT[*6] models long-range molecular interactions for property prediction, while UniteFormer focuses on combinatorial path generation and input-modality adaptation, not regression or graph-level inference.
>
> Although there may be surface-level architectural similarities (e.g., use of attention or GNNs), UniteFormer is fundamentally designed for path construction under modality-variant inputs, which is non-trivival and crucial for real-world VRPs. To our knowledge, UniteFormer is the first unified model to support all three modalities within a single framework. Each component is purposefully designed to handle node, edge, and hybrid in a scalable and flexible manner, combining the strengths of GNNs (for edge reasoning) and attention mechanisms (for node embedding). However, we agree that referencing broader literature like [*1]–[*6] will strengthen the generality and novelty of our method.
> > **Q1:Existing methods inadequate?**
>
> **A1: (1) The core limitation of existing methods** lies not in the use of masked inputs, but in their reliance on a single input modality, i.e., node-only, edge-only, or hybrid, making them less adaptable to real-world scenarios with missing or varying input types. For example: (i)Edge-only methods like MatNet may underutilize node coordinates even when available; (ii)Node-only methods such as POMO and LEHD cannot directly process edge input without redesign and retraining; (iii)Models like GCN-BS or DAR, built for hybrid inputs, perform poorly when only one modality is present due to their architectural dependence on both. To our knowledge, no existing work offers a unified framework that flexibly handles node-only, edge-only, and hybrid inputs within one model, despite its clear practical value. This motivates our design of a unified architecture that adapts to all modalities without the need for retraining.
>
> **(2) Regarding your point on** GNNs learning to internally complement node and edge features: we agree in principle. In fact, our model includes GCNs as a core component, precisely to capture structural dependencies from edge and node. However, through extensive experiments and ablation studies, we find that GCNs alone struggle to perform consistently across all input modalities, especially when only node or edge data is provided. To address this, we design a parallel mixed encoder combining GCNs and attention, enabling effective capture of both node- and edge-level semantics. This design ensures robust performance across all input modalities, as evidenced by:
> (i)Quantitative comparisons in Table 1,
> (ii)Component-level ablations in Section 4.2, and
> (iii)Additional targeted ablations (see A3).
> >**Q2:Other Tasks and Three Inputs**
>
> **A2: (1)** VRPs are a core class of combinatorial optimization problems (COPs) with widespread applications in logistics and transportation. Given their practical relevance and structural complexity, developing specialized and efficient models for VRPs is both common and well-justified in existing literature, such as LEHD, GCN-BS, DAR, ICAM, focusing solely on VRPs. Our work follows this tradition, ensuring fair comparisons with established baselines. Moreover, our model further extends this research by supporting all three input modalities, offering greater flexibility than those prior single-modality methods. Moreover, UniteFormer can also handle asymmetric problems like ATSP, which node-only models such as POMO and LEHD cannot directly address.
>
> However, UniteFormer's architecture is inherently general and we extend it to the **0-1 Knapsack Problem(KP)**. The results below for KP50 with 200 training epochs on both POMO and UniteFormer show that UniteFormer maintains impressive performance beyond VRPs. This further confirms its potential of UniteFormer as a general-purpose neural solver for other tasks beyond VRPs.
> |KP50|Optimal|POMO|UniteFormer|
> |-|-|-|-|
> |score|20.127|20.113|20.120|
> |gap|--|0.069%|**0.029%**|
>
> **(2)** While prior methods like POMO is restricted to a single input modality and require architectural changes to accommodate others, real-world VRPs demand greater flexibility. Node-based inputs are common in Euclidean settings, edge-based inputs are essential for non-Euclidean or asymmetric tasks like ATSP, and hybrid inputs combining coordinates and distances are increasingly important in practical logistics scenarios. UniteFormer is the first framework, to our knowledge, that supports all three modalities, i.e., node, edge, and hybrid, within a unified architecture, reducing the need for retraining. As shown in Table 1 and Section 4.2, it delivers strong and consistent performance across all input types. This modality flexibility is crucial in real-world deployments, e.g., node coordinates may be unavailable or noisy, while edge-level data is often more reliable.
> >**Q3:Structural Rationality**
>
> **A3:** First, to clarify, our architecture includes both GCNs and attention, and there is no absence of GCNs. So we guess what you mean by this sentence is: "If attention alone... then the inclusion of a GCN seems odd."
>
> **(1) Complementarity:**
> As described in Section 3.2, we adopt a parallel GCN-attention encoder to jointly process node and edge features and capture their cross-modal interactions. While attention mechanisms (e.g., POMO, LEHD) are effective for node-centric representations, GCNs are well-suited for learning from structured graph data, especially when edge input is available or critical. This hybridization is particularly beneficial in tasks like CVRP and ATSP, where edge weights (e.g., asymmetries, capacities) encode essential semantics that attention alone may overlook. In fact, Section 3.3(lines 157–161) further explains that attention modules are introduced to complement the limited node-level expressiveness of GCNs, ensuring a balanced and expressive node embedding.
>
> **(2) Input Modality:**
> UniteFormer is designed to handle three input modalities. Given this, attention and GCNs serve distinct yet complementary roles: GCNs are naturally suited for Edge-only and hybrid inputs, enabling relational reasoning over graphs. Attention excels in processing Node-only data, especially when no explicit graph structure is available. Empirically, attention-only methods like POMO work well for Node-only inputs but perform poorly when edge features dominate. As shown in Table 1, our variant "POMO-edge"(a modified POMO model applied to edge-based input) performs notably worse than UniteFormer. This confirms that attention alone is insufficient for edge-dominant tasks. Moreover, in problems like ATSP, where edge asymmetry is key and node coordinates cannot infer distance, GCN-based reasoning is indispensable. As demonstrated in Appendix F, our hybrid model surpasses attention-only methods like MatNet on ATSP.
>
> **(3) Empirical Validation:**
> To validate our architectural design, we conducted **four additional ablation studies** (see response A4 to Reviewer HhBh), examining (i) the complete removal of the GCNs module and (ii) reordering by placing self-attention before GCNs. As shown in the table below, both changes led to noticeable performance drops, underscoring the necessity of GCN and the soundness of our original design.
>
> In addition, we further conducted ablation studies on the mixed-score attention and parallel encoding structures (see the table below), confirming their effectiveness, particularly for edge-only inputs. Our initial submission already included ablation results for three architectural components (see Appendix E), further demonstrating the value of self-attention and the overall structure.
> |TSP50-Gap%|w.o.GCN|w.o.Order|w.o.mixeMHA|w.o.decoding|w.o.self-attention| w.o.node-focused|w.o.edge-aware|UniteFormer|
> |-|-|-|-|-|-|-|-|-|
> |input-E|3.261|0.074|123.029|25.055|0.007|0.004|1.294|0.004|
> |input-X|0.616|6.735|0.192|0.003|0.009|0.004|0.006|0.004|
> |input-XE|0.616|0.069|0.129|0.003|0.004|0.003|0.005|0.003|
>
> |CVRP50-Gap%|w.o.GCN|w.o.Order|w.o.mixeMHA|w.o.decoding|w.o.self-attention| w.o.node-focused|w.o.edge-aware|UniteFormer|
> |-|-|-|-|-|-|-|-|-|
> |input-E|3.467|45.704|100.430|2.271|0.770|0.841|37.266|0.668|
> |input-X|0.936|31.186|0.801|0.820|0.721|1.103|1.835|0.694|
> |input-XE|0.936|18.521|0.799|0.785|0.680|1.028|1.734|0.660|

---

> > ### Author Response · Authors · 2025-08-05
> >
> > Reviewer Fawv,
> >
> > We sincerely appreciate your constructive and thoughtful comments, especially for bringing literatures of other domains to our attention. As the author-reviewer discussion period is ending soon, could you please kindly take a look at our response at your early convenience, so that we will have enough time to address the remaining concern if any. Thanks for your understanding and support.
> >
> > Best regards,
> > Authors

---

> > > ### Comment · Reviewer_Fawv · 2025-08-06
> > >
> > > Thank you, and I apologize for the delayed response.
> > >
> > > First, I would like to thank the authors sincerely for their thorough and thoughtful rebuttal. Each of the questions I raised was addressed in detail. In particular, I appreciate the clarification regarding question 3, which resolved a misunderstanding on my part.
> > >
> > > As I am not deeply familiar with the history of VRP research, I cannot definitively assess whether the proposed architecture is novel in that context. However, based on the authors' explanation, I understand that it is an original contribution to the field of VRP. From my perspective, the architecture appears to be a well-designed integration of existing techniques. Initially, I interpreted it as a case study of such an integration applied to a specific application rather than as a fundamentally new architecture. However, I recognize that my position as a non-specialist in VRP literature may limit this interpretation.
> > >
> > > Taking into account the responses to my concerns, as well as the discussions and evaluations provided by other reviewers, I agree with the positive overall sentiment and am inclined to support acceptance.

---

> > > > ### Author Response · Authors · 2025-08-06
> > > >
> > > > We sincerely thank the reviewer for acknowledging our response and supporting our work. We are especially appreciative of the insightful comments and perspectives from other domains, which have helped us better position our contributions and claims. We thank the reviewer again for raising the score!

---

### Official Review · Reviewer_HhBh · 2025-07-01

**Clarity:** 3
**Significance:** 2
**Originality:** 3
**Rating:** 5
**Confidence:** 4

**Summary:**

The paper proposes a novel neural network architecture that can take different kinds of inputs (edge only, node only, or both) for solving routing problems.

**Questions:**

1. What's the model size of UniteFormer compared to the other neural solvers used as baselines?

2. On which instance size UniteFormer is trained on? For instance, is it trained on TSP-n and then evaluated on TSP-n?

**Ethical Concerns:**

["NO or VERY MINOR ethics concerns only"]

**Final Justification:**

The authors have address the weaknesses I mentioned. Although the size of the proposed model is large, it's comparable to other ones. In addition, its generalization capability seems to be good.

**Limitations:**

yes

**Quality:**

3

**Strengths And Weaknesses:**

The idea of training a unified network to be accept different inputs is novel as far as I know. The trained solvers seem to be robust and have excellent performance.

The weaknesses seem to be larger model size and possible generalization issues.

The presentation of the neural architecture is not explained in sufficient details, e.g.,

- The mixed-scoreMHA layer is not explained properly, even in Appendix A. For instance, it's not clear where h_0 and h_v are used in Appendix A.
- Some notations are not explained, such as \Theta_{ij}^{knn} in Equation 4.
- The MHA function is used with different parameters: MHA(h, H) or MHA(q, k, v). Again, for completeness, the notations should be explained.

The experimental set-up is missing some details:

- What's the model size of UniteFormer compared to the other neural solvers used as baselines?
- On which instance size UniteFormer is trained on? For instance, is it trained on TSP-n and then evaluated on TSP-n?

Since the proposed architecture is relatively complex, the authors should do an ablation study to demonstrate that different components of this architecture are useful (e.g., mixed-score attention layer, parallel-attention decoding…)

Minor remarks:

In line 20, "These methods learn solution strategies autoregressively from data": This sentence seems to ignore all the machine learning based methods used as improvement solvers.

In line 41: "capture complementary structural information": In Euclidean TSP, the node positions and the distances seem to include the same information. What do you mean by "complementary"?

In line 115: "node features are passed to the GCN directly": In Fig. 1, there's no arrow from node embedding to GCN. Is it normal?

---

> ### Author Rebuttal · Authors · 2025-07-31
>
> Thank you for your constructive and valuable comments and suggestions. We address your concerns point-by-point as follows:
>
> > **Q1: Notation Explanation.**
>
> **A1:** We appreciate your concern regarding the clarity of the **mixed-score MHA layer** and the associated notation in Appendix A. In the main text, the mixed-score MHA layer is briefly introduced by listing the required input arguments, specifically, $\text{mixed-scoreMHA}(h_0,h_v,D_{ij})$, to emphasize the functional interface rather than the internal details. A more detailed explanation of this mechanism has already been provided in **Appendix A.1**, particularly in Equation 1, where the general input embedding is denoted as $h_i$. In our implementation, $h_i$ corresponds to: a zero vector $h_0$, and a randomly selected one-hot vector $h_v$. These two are used to construct the inputs to the MHA layer as follows:
> $$q = W^Q h_0,\quad k = W^K h_v,\quad v = W^V h_v.$$
> We opted for generic notation in the appendix to reflect the flexibility of $h_i$ as a general embedding, without tying it to specific choices like $h_0$ and $h_v$, which are task- and design-specific. That said, we agree that clarifying this correspondence more explicitly would improve readability, and we will incorporate this clarification in the final version.
>
> **Regarding your other points:**
>
> **(1) The notation $\Theta_{ij}^{knn}$** refers to the edge adjacency matrix derived from the KNN graph structure. Although this has already been mentioned in line 139 of the main text, we acknowledge that it still lacks a detailed explanation. We will provide a more detailed description in the final version to improve clarity and completeness.
>
> **(2) The notation $\text{MHA}(h,H)$** is a shorthand representation of a generic attention layer, where the output from the previous layer (e.g., hidden states) serves as both the query and context input for the next layer. In contrast, $\text{MHA}(q,k,v)$ explicitly denotes the multi-head attention computation used in the decoder stage, where the query, key, and value inputs may come from different sources. We will clearly distinguish and define these usages in the final version to avoid ambiguity.
>
> > **Q2: Model Size of UniteFormer.**
>
> **A2:** Thank you for your question. We provide a comparison of the model size (in terms of parameter count and storage footprint) between UniteFormer and other baseline neural solvers. Specifically, UniteFormer has approximately 4.48 million parameters, resulting in a model size of 47,287 KB, which is larger than POMO (as UniteFormer needs to deal with different types of input simultaneously), but smaller than MatNet.
> ||POMO-edge|MatNet|POMO|LEHD|GCN-BS|DAR|ICAM|UniteFormer|
> |:-:|:-:|:-:|:-:|:-:|:-:|:-:|:-:|:-:|
> |**Parameters**|1,282,304|5,601,696|1,269,760|1,431,681|11,053,802|1,269,760|2,457,088|4,478,736|
> |**Storage Footprint**|15,172KB|67,887KB|15,045KB|16,743KB|129,732KB|15,025KB|29,035KB|  47,287KB|
>
> This increase in size is due to UniteFormer’s unified architecture, which supports all three input modalities within a single model, without requiring separate training runs for each modality. In contrast, baseline methods typically require separate models to be trained independently for each input type. Moreover, our inference time remains competitive and is on par with most baseline methods, as shown in **Table 1** of the main paper.
>
> We will include a detailed table comparing parameter counts for all methods in the final version to improve transparency and reproducibility.
>
> > **Q3: Training Settings of UniteFormer.**
>
> **A3:** Following the standard practice in prior works like POMO and MatNet, our UniteFormer trained on TSP-n is mainly evaluated on the same instance size TSP-n. However, we have also included the generalization test on TSPLib and CVRPLib (i.e., trained on 100-node instances but tested on various sizes) in **Table 3 of the paper**. To further demonstrate strong generalization to larger problem sizes, we apply models trained on TSP-100/CVRP-100 to solve TSP-200/CVRP-200 instances (see **the table below**, which provides the gap values under three input modalities). Here, UniteFormer still achieves competitive performance, highlighting its capacity to generalize beyond the training size.
>
> |Gap|POMO-edge|MatNet|UniteFormer-E|POMO|LEHD-greedy|UniteFormer-X|GCN-BS|DAR|ICAM|UniteFormer-XE|
> |:-:|:-:|:-:|:-:|:-:|:-:|:-:|:-:|:-:|:-:|:-:|
> |TSP100|2.010%|0.213%|**0.092%**|0.153%|0.531%|**0.099%**|1.379%|0.194%|0.212%|**0.066%**|
> |TSP200|60.017%|5.408%|**2.778%**|2.144%|**0.859%**|1.424%|48.697%|1.432%|2.146%|**1.058%**|
> |CVRP100|1.843%|3.086%|**0.810%**|0.790%|3.607%|**0.727%**|--|0.890%|1.240%|**0.662%**|
> |CVRP200|181.072%|7.648%|**2.102%**|4.866%|3.312%|**1.725%**|--|2.627%|4.925%|**1.420%**|
>
> > **Q4: Ablation Study.**
>
> **A4:** Thank you for your valuable suggestion. As part of our original submission, we have already conducted ablation studies on three input modalities and three major architectural components, as detailed in **Section 4.2** and **Appendix E**. These experiments were selected to highlight the most representative design choices under the space constraints of the main paper.
>
> Following your suggestion, we have further performed **four additional ablation studies** to assess the impact of the following components:
>
> * Removing the **mixed-score attention layer** (denoted by w.o. mixeMHA),
> * Changing the **parallel-attention decoding** to a single-layer decoding architecture (denoted by w.o. decoding),
> * Removing the **GCN layer** (denoted by w.o. GCN), and
> * Placing the **self-attention layer before the GCN module** (denoted by w.o. Order).
>
> The experimental results are summarized in the table below and further confirm that each of these components contributes meaningfully to the model’s performance.
>
> |TSP50|w.o.mixeMHA|w.o.decoding|w.o.GCN|w.o.Order|UniteFormer|
> |-|:-:|:-:|:-:|:-:|:-:|
> |input-edge|123.029%|25.055%|3.261%|0.074%|**0.004%**|
> |input-node|0.192%|**0.003%**|0.616%|6.735%|0.004%|
> |input-XE|0.129%|0.003%|0.616%|0.069%|**0.003%**|
>
> |CVRP50|w.o.mixeMHA| w.o.decoding|w.o.GCN|w.o.Order|UniteFormer|
> |-|:-:|:-:|:-:|:-:|:-:|
> |input-edge|100.430%|2.271%|3.467%|45.704%|**0.668%**|
> |input-node|0.801%|0.820%|0.936%|31.186%|**0.694%**|
> |input-XE|0.799%|0.785%|0.936%|18.521%|**0.660%**|
>
> We appreciate your feedback and will include these new ablation studies in the final version to provide a more comprehensive analysis of the architecture.
>
> > **Q5: Minor remarks [line 20]: Improvement solvers.**
>
> **A5:** Thank you for the valuable comment. You're right that there exists a broad class of machine learning–based methods that function as improvement solvers (e.g., learning local search heuristics or integrating with classical optimization frameworks), which are not necessarily autoregressive in nature.
>
> Our original sentence was indeed narrow in scope. We have revised it as follows to better reflect the diversity of neural methods for VRPs:
>
> > These methods include both autoregressive models that learn construction policies from data[21, 9, 25], as well as learning-based improvement solvers that enhance classical optimization procedures. However, many of these models, particularly construction-based ones, make an overly simplifying assumption: they rely solely on either node coordinates or edge distances as input.
>
> > **Q6: Minor remarks [line 41]: Complementary.**
>
> **A6:** Thank you for your insightful question. While node positions in Euclidean TSP do determine pairwise distances, we use the term "**complementary structural information**" to highlight that our model learns distinct yet mutually reinforcing aspects of the problem via separate encoders.
>
> Specifically, the edge-aware sub-encoder directly models pairwise distances, which are critical for evaluating tour costs and capturing local geometric structure. In contrast, the node-focused sub-encoder uses attention to extract global, node-centric patterns, such as a node’s relative position in the graph (e.g., centrality, clustering). Although both sub-encoders ultimately rely on the same raw coordinates, they capture complementary views: one encodes pairwise relational structure, while the other emphasizes global positional context.
>
> In addition, although distances can be derived from coordinates in Euclidean space, we find that neural networks benefit from directly learning the true distance values rather than relying on implicitly inferred ones. This direct modeling of edge-level information leads to more effective learning and improved solution quality.
>
> Moreover, the notion of complementary becomes even more pronounced in non-Euclidean settings, such as the ATSP (the results are presented in **Appendix F**), where distances cannot be solely inferred from coordinates. This demonstrates that our model provides a flexible interface for directly handling real-world problems where true pairwise distances are explicitly defined and cannot be inferred from positions alone.
>
> > **Q7: Minor remarks [line 115]: Figure.**
>
> **A7:** Thank you for your careful observation. The sentence in line 115-"Without edge weights, node features are passed to the GCNs directly" is specifically referring to the Node-only input modality illustrated in **Figure 2(b)**. In this subfigure, the arrows clearly indicate the direct flow of node embeddings into the GCNs under the Node-only setting, and are intended to highlight the different input modes handled by our framework. In contrast, **Figure 1** is designed to provide a high-level overview of the model architecture, emphasizing the modular composition and interactions between the main components. It does not explicitly depict the detailed flow of inputs for the three input modalities, which are instead elaborated in Figure 2. As image updates are not permitted during the rebuttal phase, we will revise Figure 1 in the final version to better reflect these input pathways and avoid confusion.

---

> > ### Comment · Reviewer_HhBh · 2025-08-07
> >
> > Thank you for the explanation and the additional experiments, which mostly address my concerns. I will update my score.

---

> > > ### Author Response · Authors · 2025-08-07
> > >
> > > We sincerely appreciate the reviewer for acknowledging our response and raising the score!

---

### Official Review · Reviewer_fzsN · 2025-07-03

**Clarity:** 3
**Significance:** 3
**Originality:** 3
**Rating:** 5
**Confidence:** 4

**Summary:**

This manuscript introduces UniteFormer, a novel neural solver designed to address the Vehicle Routing Problem (VRP) by integrating both node and edge modalities within a single model. This approach aims to enhance the flexibility and generalization of neural solvers in real-world scenarios by supporting node-only, edge-only, and hybrid input types.

**Questions:**

My concerns refer to weaknesses.

**Ethical Concerns:**

["NO or VERY MINOR ethics concerns only"]

**Final Justification:**

The authors have addressed my concerns.

**Limitations:**

yes

**Quality:**

3

**Strengths And Weaknesses:**

**Strengths**

- **S1.** The paper presents a unified approach to solving VRPs by integrating multiple input modalities. The mixed encoder integrates graph convolutional networks (GCNs) and attention mechanisms to process both node and edge features collaboratively. The decoder's parallel architecture and nonlinear query mechanisms enhance the model's representational capacity.

- **S2.** The experimental results demonstrate that UniteFormer achieves state-of-the-art performance on both TSP and CVRP tasks across multiple input modalities.

**Weaknesses**

- **W1.** The manuscript lacks relevant computational complexity analysis and efficiency experiments. The heavy encoder architecture results in high computational demands, which can limit the model's scalability to very large problem instances. It might be a concern for practical applications where resources are limited.

- **W2.** The manuscript lacks a problem description and formalization. While the paper briefly mentions the setup for TSP and CVRP in the experimental section, it lacks a detailed description and formalization of the problems. This omission might make it difficult for readers to grasp how the model aligns with the specific requirements of the problem.

- **W3.** The manuscript fails to analyze hyperparameter sensitivity. It can limit the model's practical application and scalability, as readers cannot evaluate the model's stability and reliability across various scenarios. A hyperparameter sensitivity analysis helps researchers and practitioners understand how different hyperparameter settings affect model performance, thereby enabling the selection of optimal hyperparameter combinations.

- **W4.** The manuscript lacks publicly available code, which limits the reproducibility of the results.

---

> ### Author Rebuttal · Authors · 2025-07-27
>
> Thank you for your constructive and valuable comments and suggestions. We address your concerns point-by-point as follows:
>
> >  **Q1: Computational complexity analysis.**
>
> **A1:** Thank you for raising this important point. We acknowledge that our encoder architecture in UniteFormer might be heavier and increase computational demands, particularly for large-scale instances. This design was motivated by the need to learn rich structural representations across multiple input modalities, which has shown strong empirical benefits in terms of solution quality.
>
> However, we would like to emphasize that UniteFormer is more practical in real-world scenarios due to its unified architecture: it can handle three different input modalities within a single model. In contrast, existing methods typically require separate training for each input setting, leading to higher cumulative training cost and reduced flexibility. UniteFormer's ability to generalize across modalities without retraining helps reduce overall training overhead and model maintenance cost, making it more efficient in deployment settings.
>
> In terms of inference efficiency, our model achieves comparable inference time to baseline methods, as shown in **Table 1** of the main paper. In addition, we also explicitly acknowledge the computational trade-off in the original paper. In **Appendix G**, we have highlighted that reducing the computational footprint of UniteFormer while preserving its expressive power is an important direction for future work. We are dedicated to the design of more lightweight yet expressive encoder modules to improve scalability for large instances.
>
> We appreciate your suggestion and will incorporate a formal complexity analysis and efficiency discussion in the final version to make these trade-offs more transparent.
>
> >  **Q2: Problem Description and Formalization.**
>
> **A2:** Thank you for your valuable feedback. We have provided the problem descriptions and formalizations in the paper (see **Appendix C**). Due to space constraints in the main text, we have already included the detailed problem descriptions and formalizations for both the TSP and CVRP in **Appendix C**, and referenced them in the **Problem Setting** subsection of the experimental section (see **Section 4**).
>
> We acknowledge that a more explicit formulation in the main paper would improve clarity, especially for readers less familiar with these problems. If space permits in the final version, we will include a concise but formal problem definition in the main text to ensure that the model's alignment with the problem requirements is clearly understood.
>
> > **Q3: Hyperparameter Sensitivity.**
>
> **A3:** Thank you for your thoughtful suggestion. In this work, our primary focus is on analyzing the impact of the three input modalities and the architectural components of the proposed model. As such, we have conducted extensive ablation studies on both input types and architecture modules, which are presented in **Section 4.2** and **Appendix E**.
>
> For hyperparameter settings, we follow the configurations used in strong baselines such as POMO and MatNet to ensure fair comparison. Therefore, we did not include a detailed hyperparameter sensitivity analysis in the initial submission. However, we appreciate the importance of evaluating model robustness across different settings. In response to your comment, we conducted a sensitivity analysis on hyperparameter $K$, which controls the number of nearest neighbors used in the KNN graph for the GCN modules. We tested values of $K = 10, 20, 30, 40, 50$ on TSP50, and found that $K = 20$ yields the best performance, with relatively stable results across the range. The results are summarized in the table below.
>
> | TSP50  | K=10  | K=20  | K=30  | K=40  | K=50  |
> | --- | --- | --- | --- | --- | --- |
> | input-edge | 0.0039% | **0.0036%** | 0.0045% | 0.0055%  | 0.0046%   |
> | input-node | 0.0044% | **0.0038%** | 0.0048% | 0.0054%  | 0.0043%   |
> | input-XE | 0.0029% | **0.0026%** | 0.0033% | 0.0035%  | 0.0031%   |
>
> We will include this analysis in the final version and additionally plan to expand the hyperparameter sensitivity study to better support practical deployment and reproducibility.
>
> > **Q4: Reproducibility.**
>
> **A4:** Following common practice, we will release the full source code and all trained models upon acceptance of the paper. As external links are not permitted during the rebuttal phase, we promise to include links to the public code in the final version.

---

> > ### Comment · Reviewer_fzsN · 2025-08-05
> >
> > The authors have addressed my concerns. I will raise my score of this paper to 5.

---

> > > ### Author Response · Authors · 2025-08-05
> > >
> > > We sincerely appreciate the reviewer for acknowledging our rebuttal and raising the score.

---

### Decision · Program_Chairs · 2025-09-17

**Decision:**

Accept (spotlight)

**Comment:**

The paper received four ratings of 5 for acceptance from the reviewers. This paper presents a new neural solver for Vehicle Routing Problems (VRPs) that integrates node and edge modalities with state-of-the-art performance across various tasks. The approach has a well-designed architecture that enhances flexibility and generalization. There were some concerns regarding the computational complexity analysis and hyperparameter sensitivity, but the authors have addressed reviewers' concerns during the rebuttal by clarifying the model's efficiency and providing additional analyses. The paper has significant contributions in neural combinatorial optimization strong experimental results. Given the unanimous support from the reviewers, the AC recommends acceptance.